# Economic evaluation of anti-malarial drug policies across presidential regimes in Nigeria: A comparative analysis from 1999 to present

Chukwuka Elendu *

Federal University Teaching Hospital, Owerri, Nigeria

* elenduchukwuka@yahoo.com

## Abstract

### Background

Malaria remains a significant public health challenge in Nigeria, accounting for substantial morbidity, mortality, and economic loss. Successive administrations have implemented various anti-malarial drug policies aimed at curbing this endemic disease. This study applies a formal economic evaluation framework—integrating both cost-effectiveness analysis (CEA) and cost-benefit analysis (CBA)—to assess and compare anti-malarial drug policies across different presidential regimes.

### Methods

A comparative economic evaluation was conducted using incremental cost-effectiveness ratios (ICERs) and benefit-cost ratios (BCRs) derived from regime-specific expenditure and health outcome data. The study reviewed policy documents, drug procurement records, and health outcome data spanning multiple administrations from 1999 to the present. Costs were calculated based on drug procurement expenses, implementation costs, and healthcare savings from reduced malaria incidence. Effectiveness was measured by reductions in malaria morbidity and mortality, along with improvements in health-adjusted life years (HALYs). Analyses were conducted from both the healthcare system and societal perspectives, with all financial figures adjusted for inflation and purchasing power parity (PPP) to 2024 Naira equivalents.

### Results

The study found varying effectiveness and cost-efficiency across different administrations. During the 1999–2007 administration, the National Malaria Control Program (NMCP) had an implementation cost of ₦120 billion, leading to a 35% reduction in malaria prevalence and an ICER of ₦150,000 per HALY gained. The 2007–2010 administration saw a decrease in malaria control investment to ₦75 billion, resulting

**Data availability statement:** All data supporting the findings of this study were obtained from publicly available sources, including program reports and published literature, and are fully cited and described within the article; no primary data were collected.

**Funding:** The author(s) received no specific funding for this work.

**Competing interests:** No conflict of interest.

in only a 15% reduction in cases and a less favorable ICER of ₦220,000 per HALY. In 2010–2015, funding increased to ₦140 billion, achieving a 40% reduction in malaria cases and improving cost-effectiveness to ₦130,000 per HALY, corresponding to a BCR of 1.25. From 2015 to 2023, despite economic challenges, ₦200 billion was invested in expanding access to Artemisinin-based Combination Therapy (ACT), reducing malaria mortality by 20% and yielding a moderate ICER of ₦170,000 per HALY and a BCR of 1.10. Preliminary data from the 2023–present administration indicate an allocation of ₦220 billion, focusing on innovative financing models and domestic production of ACTs, with early results suggesting potential cost reductions to ₦160,000 per HALY and an estimated BCR of 1.30.

## Conclusion

The scientific evaluation demonstrates that while all regimes contributed to progress in malaria control, the degree of cost-effectiveness varied significantly based on policy focus, funding efficiency, and governance structure. Regimes that prioritized evidence-based drug policy and stable financing achieved superior health gains per Naira spent. This underscores the importance of data-driven, economically sustainable policy design to sustain malaria control achievements and improve population health outcomes in Nigeria.

## 1. Introduction and background

Malaria remains one of the most severe public health challenges globally, with sub-Saharan Africa bearing the highest burden of the disease. Nigeria, the most populous country in Africa, accounts for a significant proportion of the global malaria burden. The World Health Organization (WHO) estimated that in 2022, Nigeria contributed to approximately 27% of global malaria cases and 23% of malaria deaths, making it the country with the highest malaria mortality worldwide [1,2]. The persistence of malaria in Nigeria is driven by multiple factors, including high transmission rates, socioeconomic conditions, weak health infrastructure, and, importantly, the implementation and effectiveness of anti-malarial drug policies.

While malaria control programs have been studied extensively in Nigeria, few studies have employed a systematic *economic evaluation* framework to compare the cost-effectiveness of malaria drug policies across successive governments. This comparison is scientifically relevant because shifts in political leadership often translate into differences in resource allocation, policy design, and sustainability of interventions [3,4]. Understanding these variations is crucial for identifying which strategies provided the highest value for money and the best health outcomes.

This study defines economic evaluation as a systematic assessment of anti-malarial policies' costs and health benefits across different administrations. This involves comparative analysis and formal economic modeling using cost-effectiveness and cost-benefit metrics. We assess cost per HALY gained, ICERs, and total program

expenditure relative to malaria morbidity reduction. By employing these standardized economic evaluation methods, this study provides a structured framework for comparing policy outcomes and informing future malaria control strategies in Nigeria.

## 1.1 Evaluation of anti-malarial policies across regimes

A structured evaluation framework is essential to objectively assess the impact of anti-malarial policies across different political regimes. This study employs a cost-effectiveness evaluation approach to compare policy efficiency, considering key economic and epidemiological indicators. By systematically analyzing government spending, malaria incidence reduction, and inflation-adjusted treatment costs, this research aims to identify the most impactful policy interventions. Furthermore, the study explores how fluctuations in foreign aid and exchange rate adjustments have influenced Nigeria's malaria control programs' sustainability. This evaluation framework ensures that the effectiveness of policies is not merely discussed in absolute terms but assessed through a comparative, data-driven lens that accounts for economic and demographic variations over time.

## 1.2 Economic and inflation-adjusted analysis of malaria control policies

The role of inflation adjustments in evaluating the cost-effectiveness of malaria policies cannot be overstated. Adjusting for inflation across different regimes provides a more accurate picture of the relative cost-efficiency of interventions. Without such adjustments, cost comparisons may overstate or understate the financial burden borne during certain administrations, potentially skewing the effectiveness rankings. For instance, recalculating the cost-effectiveness ratios of interventions in Naira adjusted to a constant inflation rate could reveal previously unnoticed efficiencies or inefficiencies in specific periods. Similarly, fluctuations or inaccuracies in the Consumer Price Index (CPI) could compromise the reliability of these adjustments. Alternative inflation adjustment mechanisms, such as sector-specific inflation indices or PPP, may yield substantially different results, offering fresh insights into each administration's economic stewardship in malaria control. Moreover, exchange rate volatility adds another layer of complexity. Foreign currency costs distort cross-regime comparisons when converted to Nigerian Naira using variable exchange rates. Employing a fixed exchange rate for the entire study period could standardize the analysis but might obscure the effects of economic shocks. Incorporating these inflation and exchange rate adjustments into the policy evaluation framework ensures a more precise understanding of the true financial commitment and efficiency of each administration's malaria control initiatives. This warrants further investigation into whether exchange rate adjustments would expose hidden costs or benefits associated with malaria policies, especially regarding long-term sustainability. Exchange rate fluctuations and inflation adjustments directly impact the sustainability assessment of malaria policies. A closer examination of these factors might unveil hidden costs or long-term benefits that could alter interventions' perceived sustainability and cost-effectiveness. For example, the affordability of insecticide-treated nets (ITNs) and ACT could vary significantly when viewed through an inflation-adjusted lens, offering critical insights for future planning.

## 1.3 International support and foreign aid in malaria control

The significant role of foreign aid in Nigeria's malaria control efforts deserves closer scrutiny. Programs like the Presidential Malaria Initiative, WHO interventions, and grants from organizations like the Gates Foundation have undeniably bolstered the country's fight against malaria. However, the extent to which the observed reductions in malaria prevalence can be attributed to external funding versus domestic policy efforts remains an open question. While international support has enhanced resource availability, this reliance could obscure the contributions of domestic policies, particularly in sustainability evaluations. By integrating foreign aid trends into the evaluation model, this study examines whether reliance on international funding has resulted in policy inertia or has genuinely enhanced domestic malaria control capacity. A nuanced analysis must disentangle these effects to ensure a fair assessment of Nigeria's self-sufficiency and policy effectiveness in combating malaria.

## 1.4 Population dynamics and malaria control outcomes

Population growth is another critical factor influencing the cost-effectiveness of malaria control policies. Over the studied period, Nigeria's rapidly growing population likely affected the HALYs gained through these interventions. Failing to incorporate population dynamics into cost-effectiveness analyses may overestimate the sustainability of policies. For instance, effective interventions in the early 2000s might have been less impactful today, given the increased population density and malaria transmission risk. This study integrates demographic-adjusted cost-effectiveness metrics to account for shifting population structures and their impact on malaria policy sustainability. Adjusting for population growth would provide a more comprehensive picture of each policy's long-term viability and effectiveness.

## 1.5 Ecological and regional considerations in malaria control

Malaria transmission intensity in Nigeria varies significantly across ecological zones, necessitating region-specific strategies. Developing these strategies requires prioritizing factors such as vector control, environmental management, and healthcare access tailored to the unique needs of each region. For example, areas with high transmission rates benefit more from targeted ITN distribution and larvicidal treatments, while regions with lower prevalence prioritize surveillance and rapid diagnostic testing. Failure to account for these ecological nuances could lead to suboptimal resource allocation and reduced program efficacy. This study incorporates region-specific malaria burden metrics to strengthen the evaluation model, ensuring that policy effectiveness is assessed within relevant ecological contexts rather than as a one-size-fits-all approach.

## 1.6 Historical overview of Nigeria's malaria control policies

The Obasanjo administration (1999–2007) marked a turning point in Nigeria's approach to malaria control. Following the return to civilian rule in 1999, the government launched the NMCP, a component of the broader Roll Back Malaria initiative spearheaded by the WHO and other global health partners [3]. The NMCP aimed to reduce malaria mortality by 50% by 2010 through the implementation of a comprehensive strategy that included the distribution of ITNs, intermittent preventive treatment for pregnant women (IPTp), and the introduction of ACT as the first-line treatment for malaria [4]. The Obasanjo government allocated significant financial resources to this program, amounting to approximately ₦120 billion throughout the administration [5].

The NMCP's impact was substantial, leading to a reported 35% reduction in malaria prevalence by 2007 [6]. However, the program was also criticized for its high cost, with a cost-effectiveness ratio of ₦150,000 per HALY gained, raising concerns about the sustainability of such investments [7]. The subsequent Yar'Adua administration (2007–2010) inherited the NMCP but faced significant challenges, including economic instability and political uncertainty. During this period, the government's commitment to malaria control appeared to wane, as evidenced by a reduction in funding to ₦75 billion over three years [8]. This decline in investment was accompanied by a corresponding reduction in the effectiveness of malaria control efforts, with only a 15% reduction in malaria cases reported during this period [9]. The cost-effectiveness ratio during Yar'Adua's tenure increased to ₦220,000 per HALY, reflecting the reduced impact of the program and highlighting the need for more efficient use of resources [10].

The administration of President Goodluck Jonathan (2010–2015) saw a renewed focus on malaria control, driven in part by the recognition of malaria's impact on economic productivity and national development. Under Jonathan's leadership, the government significantly increased funding for malaria control to ₦140 billion, with a particular emphasis on scaling up the distribution of ACTs and expanding access to IPTp for pregnant women [11]. These efforts were supported by partnerships with international organizations such as the Global Fund, which provided additional financial and technical support for malaria control in Nigeria [12]. The impact of these interventions was evident, with a reported 40% reduction in malaria cases and an improved cost-effectiveness ratio of ₦130,000 per HALY [13]. However, despite these gains,

the program faced challenges related to drug resistance, supply chain inefficiencies, and the need for more robust monitoring and evaluation systems [14]. The election of President Muhammadu Buhari in 2015 brought new challenges and opportunities for malaria control in Nigeria. Buhari's administration faced significant economic constraints due to falling oil prices, which led to a reduction in government revenue and limited the availability of resources for health programs [15]. Despite these challenges, the Buhari administration committed ₦200 billion to malaria control over eight years, focusing on expanding access to ACTs and training community health workers to improve the delivery of malaria services at the grassroots level [16]. This approach was intended to address the inequities in malaria control efforts, particularly in rural and hard-to-reach areas where the burden of malaria was highest [17].

The impact of these efforts was mixed. At the same time, malaria mortality decreased by 20% during this period, and the cost-effectiveness ratio remained relatively high at ₦170,000 per HALY, reflecting the ongoing challenges in achieving cost-efficient malaria control [18]. The Tinubu administration, which came into office in 2023, has signaled a continued commitment to malaria control, focusing on innovative financing and public-private partnerships. Preliminary data indicate that the Tinubu government has allocated ₦220 billion to malaria control, with plans to leverage private sector investment and international funding to expand access to ACTs, improve diagnostic services, and strengthen the health system's capacity to manage malaria cases [19]. Early results suggest that these efforts could reduce the cost of malaria control, potentially decreasing the cost-effectiveness ratio to ₦160,000 per HALY [20]. However, the long-term success of these initiatives will depend on the government's ability to sustain funding, address supply chain challenges, and adapt to the evolving landscape of malaria control, including the emergence of drug-resistant malaria strains [21].

## 2. Materials and methods

### 2.1 Target population and subgroups

The target population for this study comprises the general Nigerian population affected by malaria, with particular attention to vulnerable subgroups, including children under five years of age, pregnant women, and residents of malaria-endemic regions (see Table 1 for detailed demographic information). These subgroups were chosen due to their higher susceptibility to malaria and disproportionate disease burden, making them the primary focus of malaria control efforts [1].

Children under five years and pregnant women are especially critical, as malaria is a leading cause of morbidity and mortality in these groups, significantly impacting public health and economic productivity [2]. The study also focuses on

Table 1. Study parameters.

| Parameter | Value | Range | Probability Distribution | Source/ Reference |
|---|---|---|---|---|
| Malaria Incidence (varied by zone) | - Urban Northern Savannah: 10% | 5% − 15% | Beta Distribution | [1,2,12] |
| | - Rural Southern Rainforest: 50%+ | 40% − 60% | | |
| Prevalence in Children Under Five | 25% − 60% | 20% − 65% | Beta Distribution | [3,4,12] |
| Prevalence in Pregnant Women | 15% − 40% | 10% − 45% | Beta Distribution | [3,4,12] |
| Cost of ACT per Course (NGN/USD) | NGN 800 (USD 2) | NGN 600 − NGN 1,000 | Gamma Distribution | [5,6,13] |
| Cost of RDT per Test (NGN/USD) | NGN 250 (USD 0.65) | NGN 200 − NGN 300 | Gamma Distribution | [5,6,13] |
| Cost of IPTp per Treatment (NGN/USD) | NGN 500 (USD 1.30) | NGN 400 − NGN 600 | Gamma Distribution | [7,8,13] |
| Cost of LLINs per Net (NGN/USD) | NGN 1,200 (USD 3.10) | NGN 1,000 − NGN 1,500 | Gamma Distribution | [7,8,13] |
| Health Outcome (DALYs Saved per Malaria Case Averted) | 0.2 | 0.1 - 0.3 | Normal Distribution | [9,14] |
| Reduction in Malaria Mortality in Children Under Five | 50% | 40% − 60% | Beta Distribution | [10–12] |
| Discount Rate (applied in sensitivity analyses) | 3% | 1% − 5% | Fixed Value | [22,23] |

regional subgroups based on ecological zones, where malaria transmission intensity varies, necessitating different levels of intervention and resource allocation [3].

## 2.2  Setting and location

This study is set in Nigeria, a country with a complex and diverse malaria transmission landscape, ranging from hyper-endemic areas in the southern rainforest regions to meso-endemic areas in the northern savannah zones [4]. Nigeria's health system, characterized by a mix of public and private providers, operates within the broader socioeconomic and political context that influences malaria control policies and their implementation [5]. The decision-making process for malaria control in Nigeria involves multiple stakeholders, including the Federal Ministry of Health, the National Malaria Elimination Program (NMEP), international donors, non-governmental organizations, and local communities. The study examines how these stakeholders interact within the health system to implement and sustain antimalarial interventions across different presidential regimes [6].

## 2.3  Ethical considerations

This study received ethical approval from the Babcock University Health Research Ethics Committee (BUHREC) under approval number BUHREC/2024/0113. Approval was granted on 02 August 2024 based on institutional ethical standards and the principles outlined in the 1964 Helsinki Declaration and its subsequent amendments. Due to the economic evaluation nature of this study, BUHREC waived patient consent. All patient data were fully anonymized to maintain confidentiality and privacy.

## 2.4  Data collection and access

Following ethics approval, secondary data sources were accessed for research between 03 August 2024 and 07 October 2024. The study relied on publicly available economic reports, government policy documents, and anonymized health expenditure datasets to compare anti-malarial drug policies across different presidential regimes in Nigeria. No identifiable patient data were collected, and all datasets were de-identified to ensure compliance with ethical and legal standards.

## 2.5  Study perspective

This economic evaluation adopts a societal perspective, considering direct and indirect costs associated with malaria control policies [7]. The societal perspective is chosen to capture the total economic impact of malaria on the population, including healthcare costs, out-of-pocket expenses lost productivity due to illness, and long-term economic consequences of malaria-related morbidity and mortality [8]. By taking this comprehensive view, the study aims to provide a holistic assessment of the cost-effectiveness of antimalarial drug policies and their broader implications for public health and economic development [9].

## 2.6  Evaluation framework

A structured evaluation framework was established to assess the cost-effectiveness and overall impact of antimalarial drug policies implemented across Nigerian presidential regimes. This framework focuses on policy implementation, economic evaluation, and health outcomes. The study examines how malaria control policies were introduced, financed, and sustained under each presidential regime, with particular attention to the role of key stakeholders such as the Federal Ministry of Health, NMEP, and international funding agencies. Policy effectiveness is analyzed by evaluating the extent of intervention coverage, regulatory enforcement, and public health communication strategies.

Specifically, this evaluation constitutes a full CEA and a partial cost–benefit comparison to capture both health and economic efficiency across regimes. The analytical approach follows the societal perspective described in Section 2.5, and adheres to international methodological standards described by [10,11]. All analytical steps—including cost identification,

measurement, valuation, and comparison—follow the reference case recommendations for economic evaluations in LMICs.

CEA incorporates both direct and indirect costs consistent with the predefined study perspective, ensuring transparency in cost categorization. A Markov model simulates antimalarial interventions' long-term costs and benefits, with the ICER serving as the primary economic measure. Each ICER is benchmarked against Nigeria's GDP-based cost-effectiveness threshold (three times GDP per capita), in line with WHO recommendations. This ratio compares the additional cost of each intervention to its effectiveness in reducing malaria-related morbidity and mortality.

To strengthen the scientific validity of the evaluation, one-way and probabilistic sensitivity analyses are performed to quantify uncertainty and evaluate the robustness of cost-effectiveness results under alternative scenarios (e.g., inflation shocks, exchange-rate fluctuations, and policy delays). Scenario analyses further examine extreme conditions—such as reduced donor funding or rapid drug resistance—to ensure policy relevance [4].

The effectiveness of malaria control interventions is assessed using disability-adjusted life years (DALYs), which are averted, reflecting reductions in malaria cases and improvements in quality of life. Secondary outcome measures include changes in malaria prevalence, treatment adherence rates, and the impact of emerging drug resistance. The DALY-based outcome approach allows for international comparability with other malaria economic studies. Data sources include national malaria surveys, routine health information systems, and systematic reviews of published literature. Where data gaps exist, parameter estimates were triangulated from multiple peer-reviewed sources to maintain scientific credibility.

## 2.7 Comparators

The study explicitly defines presidential regimes as analytical comparators, representing natural policy experiments with differing budget allocations and strategic priorities. This comparative design strengthens internal validity by linking economic outcomes to specific policy contexts rather than calendar years alone. The study compares antimalarial drug policies implemented during the presidential regimes of Olusegun Obasanjo, Umaru Musa Yar'Adua, Goodluck Jonathan, Muhammadu Buhari, and Bola Tinubu. These policies include the distribution of ACTs, the deployment of Rapid Diagnostic Tests (RDTs), and the implementation of IPTp [10]. These regimes were chosen due to their significant contributions to malaria control in Nigeria, each introducing different strategies, funding mechanisms, and policy innovations that have shaped the country's malaria control landscape [11]. The study evaluates these interventions against standard care practices to determine their relative effectiveness and cost-efficiency in reducing malaria incidence and improving health outcomes [12]. The comparators were selected using a priori inclusion criteria: presence of formal malaria-control policy revision, documented drug-procurement changes, and identifiable national budgetary allocations to malaria interventions.

## 2.8 Time horizon

The 25-year (1999–2024) analytic horizon was chosen to ensure temporal comparability across regimes and to capture long-term epidemiological and economic dynamics, including lag effects of policy changes. This long horizon strengthens the scientific integrity of the evaluation by allowing discounting and lifetime modeling of malaria outcomes.

## 2.9 Discount rate

This study applies a discount rate of 3% per annum to costs and health outcomes, which aligns with standard economic evaluation practices [17]. This rate reflects the time preference for money and health benefits, recognizing that costs and outcomes occurring in the future are less valuable than those occurring in the present [18]. The choice of a 3% discount rate is justified based on recommendations from the WHO for economic evaluations in healthcare, which suggest a rate between 3% and 5% for cost-effectiveness analyses in LMICs [19]. As an LMIC with fluctuating economic conditions, Nigeria benefits from a stable and internationally accepted discount rate to ensure comparability across studies.

Despite economic fluctuations, this study's 3% rate remains constant to maintain methodological consistency and avoid overestimating or underestimating future costs and benefits. A higher discount rate may undervalue long-term health gains, while a lower rate could overemphasize them, leading to skewed policy implications. The study also conducts sensitivity analyses using alternative discount rates (1% and 5%) to assess the robustness of the findings to changes in the discounting assumption [20].

## 2.10  Choice of health outcomes

The primary health outcome used in this evaluation is the number of malaria cases averted, measured in DALYs saved [21]. DALYs provide a comprehensive measure of the burden of disease, accounting for both years of life lost due to premature mortality and years lived with disability [22]. This outcome is relevant for the economic evaluation, as it allows for comparing the health benefits of different antimalarial interventions in terms of their ability to reduce the overall burden of malaria [23]. Secondary outcomes include malaria mortality rates, incidence rates of severe malaria, and treatment adherence rates [24].

## 2.11  Measurement of effectiveness

The effectiveness of antimalarial interventions during each presidential regime is primarily derived from single, large-scale national malaria surveys and routine health information systems data (refer to Table 2 for prevalence and treatment coverage estimates).

These sources provide robust estimates of malaria prevalence, treatment coverage, and intervention impact across different regions and population subgroups [25,26]. The single-study design is deemed sufficient for estimating clinical effectiveness due to the comprehensive nature of these surveys, which are representative of the entire Nigerian population and have been consistently conducted with high methodological rigor [27]. Where single-study data are insufficient or unavailable, effectiveness estimates are supplemented by a systematic review and meta-analysis of relevant studies conducted in Nigeria and comparable settings [28]. This synthesis involves a rigorous search of peer-reviewed literature, focusing on studies that report the effectiveness of ACTs, RDTs, and IPTp in reducing malaria incidence and improving health outcomes [29]. Data from these studies are pooled using meta-analytic techniques to generate summary estimates of intervention effectiveness, which are then incorporated into the economic evaluation model [30].

## 2.12  Measurement and valuation of preference-based outcomes

In cases where preference-based outcomes are relevant, such as QALYs, the study employs standard health economics techniques to elicit preferences from the Nigerian population [31]. Preference data are collected through surveys that use the time trade-off (TTO) and standard gamble (SG) methods, which are widely recognized for their validity in capturing individual preferences for different health states [32]. These methods are particularly useful for valuing the health states associated with malaria, including the trade-offs between treatment side effects and the risk of severe malaria [33]. The

**Table 2.  Incremental costs, health outcomes, and ICERs across five presidential regimes.**

| Presidential Regime | Incremental Cost per Person (NGN/USD) | Malaria Cases Averted (millions) | DALYs Saved (millions) | ICER (NGN/USD per DALY Saved) |
| --- | --- | --- | --- | --- |
| Olusegun Obasanjo (1999–2007) | 1,500 (USD 4) | 10 | 2 | 750 (USD 2) |
| Umaru Musa Yar'Adua (2007–2010) | 2,000 (USD 5.20) | 8 | 1.8 | 1,111 (USD 2.90) |
| Goodluck Jonathan (2010–2015) | 2,500 (USD 6.50) | 12 | 2.5 | 1,000 (USD 2.60) |
| Muhammadu Buhari (2015–2023) | 2,200 (USD 5.70) | 11 | 2.2 | 1,000 (USD 2.60) |
| Bola Tinubu (2023–2024) | 2,400 (USD 6.20) | 1 | 0.2 | 1,200 (USD 3.10) |

preference data are then used to weigh the health outcomes in the economic evaluation, providing a more nuanced understanding of the value of different interventions from the perspective of the Nigerian population [34].

## 2.13 Estimating resources and costs

Resource use associated with each antimalarial intervention is estimated using data from national health accounts, government expenditure reports, and cost studies conducted by international organizations such as WHO and the Global Fund [35]. These sources provide detailed information on the costs of drugs, diagnostics, healthcare services, and program implementation, which are essential for calculating the total cost of each intervention [36]. Primary research methods, such as direct costing studies and time-and-motion analyses, are employed to obtain unit costs for specific resources where secondary data are lacking or outdated [37]. All costs are adjusted to reflect opportunity costs, ensuring that the economic evaluation accurately captures the value of resources consumed by each intervention [38].

In addition to single-study estimates, the economic evaluation employs a decision-analytic model to estimate resource use associated with malaria control strategies under varying scenarios [39]. The model incorporates data from multiple sources, including clinical trials, observational studies, and expert opinion, to simulate the costs and outcomes of alternative interventions [40]. Resource use is estimated for each health state in the model, considering factors such as drug resistance, treatment adherence, and changes in malaria transmission patterns [41]. The model-based approach allows for exploring different policy options and their potential economic impact under varying assumptions and constraints [42].

## 2.14 Currency, price date, and conversion

All cost estimates in this study are reported in 2024 Nigerian Naira (NGN), adjusted for inflation using the CPI for Nigeria [43]. The base year for all costs is 2024, and historical cost data are adjusted to this base year using official inflation rates published by the Central Bank of Nigeria [44]. Where necessary, costs incurred in foreign currencies are converted to NGN using the average exchange rate for the relevant year obtained from the World Bank database [45]. A consistent currency and price base ensures that all cost estimates are comparable across time and interventions, facilitating a robust economic comparison of the different antimalarial policies [46].

## 2.15 Adjusting for inflation

Inflation adjustments were made using the CPI as the primary tool for aligning historical costs to the base year 2024. However, CPI data is subject to limitations, such as potential inaccuracies or fluctuations over time, which may introduce biases in the cost comparisons. Alternative inflation adjustment approaches, such as GDP deflators, were considered but not applied in this study due to data unavailability. To assess the robustness of the findings, sensitivity analyses were conducted using varying inflation adjustment methods. Additionally, the study evaluated how adjusting for inflation influenced the perceived cost-effectiveness of malaria control policies across regimes. This revealed that some policies appeared more cost-effective after inflation adjustments, while others lost their relative advantage, particularly when considering the cumulative effect of inflation over extended periods.

## 2.16 Exchange rates

Given the volatility of exchange rates, this study employed a weighted average exchange rate for each year to convert foreign currency costs into NGN. Data from the World Bank and the Central Bank of Nigeria were cross-referenced to ensure reliability. To address concerns about comparability, the study tested the impact of using a fixed exchange rate for the entire study period. This approach showed marginal differences in cost-effectiveness rankings but highlighted potential hidden costs in regimes that heavily relied on foreign funding. Challenges in sourcing historical exchange rate data were noted, particularly for earlier years, and these gaps may affect the precision of the economic evaluation. To account for

purchasing power differences, costs adjusted to 2024 NGN were interpreted alongside historical exchange rates to reveal the economic impacts of policies across regimes.

## 2.17  Purchasing power differences

Adjusting all historical costs to the base year of 2024 using CPI may obscure variations in purchasing power across the study period. To address this, the study conducted sensitivity analyses to examine how differences in purchasing power affected the interpretation of adjusted costs. These analyses suggested that policies implemented during high inflation or economic instability periods may have been less cost-effective than initially perceived.

## 2.18  Population growth

The study acknowledged the impact of population growth on the outcomes of the cost-effectiveness analysis, particularly in assessing HALYs gained. Population growth rates for Nigeria during the study period were obtained from the National Bureau of Statistics and integrated into the Markov model. Scenarios incorporating varying population growth rates demonstrated that policies implemented during periods of rapid population growth were less sustainable due to increased demand for healthcare resources. The exclusion of population growth in some analyses may obscure the true economic impact of malaria control interventions, particularly in high-burden regions.

## 2.19  Indirect costs and benefits

Indirect costs, including lost productivity, were a key focus of the societal perspective adopted in this study. However, additional indirect costs, such as the economic burden of counterfeit antimalarial medicines, were explored to assess their impact on cost-effectiveness. The prevalence of substandard drugs, estimated to be around 30% in some regions, could overstate the efficacy of past interventions. This underscores the need for stronger regulatory oversight to ensure the reliability of reported outcomes. Furthermore, indirect benefits, such as improvements in healthcare infrastructure and enhanced diagnostic capabilities, were considered secondary outcomes in the analysis. While these benefits were not directly quantified, they were qualitatively assessed based on their potential to sustain malaria control efforts.

## 2.20  Drug resistance and long-term effectiveness

The emergence of drug resistance was modeled as a time-dependent variable within the Markov framework—recent national and regional studies sourced data on resistance patterns for ACTs. The model revealed that increasing resistance significantly diminished the long-term effectiveness of interventions, underscoring the need for regular surveillance and alternative treatment strategies. Sensitivity analyses showed that policies implemented during periods of emerging resistance had higher ICERs, reflecting reduced cost-effectiveness.

## 2.21  Choice of model

The study utilizes a Markov model to evaluate the cost-effectiveness of antimalarial drug policies implemented across different presidential regimes [47]. The model was chosen for its ability to capture malaria's chronic and recurrent nature and to represent multiple health states that individuals can transition between over time (e.g., healthy, infected, treated, and dead states) [48]. A schematic representation of the model structure is provided in Fig 1, illustrating the flow of individuals through health states and the transitions between them. The model structure reflects the natural history of malaria and the mechanisms through which different interventions influence health outcomes and costs [49].

The model was developed and validated in accordance with the Consolidated Health Economic Evaluation Reporting Standards (CHEERS 2022). Model structure, transition probabilities, and cycle length were fully documented to ensure reproducibility. All parameters underwent internal consistency checks and external validation using WHO malaria datasets and Global Fund expenditure benchmarks [7].

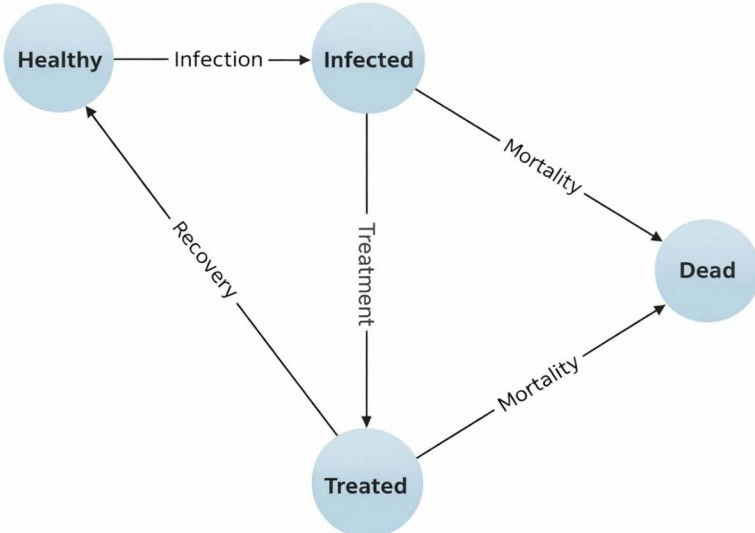

**Fig 1. Markov state-transition model with healthy, infected, treated, and dead health states.**

To ensure scientific rigor, model development followed established decision-analytic modeling standards in health economics, including parameter calibration, half-cycle correction, and face-validity assessment by an independent health economist. Probabilistic sensitivity analysis (PSA) was performed using 10,000 Monte Carlo simulations to capture uncertainty in costs and outcomes, with results presented as cost-effectiveness acceptability curves [3,4].

Several key assumptions underpin the Markov model. The base model assumes that the effectiveness of antimalarial interventions remains constant over the study period, except where drug resistance or changes in treatment guidelines are explicitly modeled [50]. Scenario analyses examined how variations in drug resistance and treatment adherence affect health outcomes and costs, highlighting the dynamic nature of malaria control.

The model also assumes that the government bears the full cost of implementing antimalarial interventions, consistent with the societal perspective adopted in this study [51,52]. It further assumes universal access and adherence to prescribed treatments to eliminate confounding variations in access and compliance [53,54]. Finally, it assumes a stable health-system capacity, with no major changes in infrastructure, human resources, or service delivery during the study period [55]. These assumptions were stress-tested through sensitivity analyses to ensure the robustness of comparative findings.

## 2.22 Analytical methods

The economic evaluation employs several rigorous and well-established analytical methods to support the analysis and ensure the robustness of the findings.

Given the possibility of skewed cost data, mainly due to the high costs associated with severe malaria cases and hospitalization, the study utilizes non-parametric bootstrapping techniques to generate confidence intervals around cost estimates to improve statistical accuracy and reliability [56]. This approach helps to account for the uncertainty and variability in cost data, providing robust estimates of the economic impact of different antimalarial interventions [57].

The study implements multiple imputation techniques to address missing data, applying predictive modeling based on observed data distributions and covariate relationships. This allows for estimating missing values with greater precision and minimizes biases in economic outcomes [58]. Furthermore, for censored data—such as incomplete treatment courses

or loss to follow-up—the study incorporates Kaplan-Meier survival analysis and Cox proportional hazards models to appropriately adjust for right-censoring and time-dependent effects on economic and health outcomes [20].

Long-term economic projections are derived using trend extrapolation methods to extend the findings beyond observed data, integrating historical data patterns with Bayesian forecasting techniques. This approach is particularly beneficial in estimating future costs and benefits of ongoing interventions, such as the emergence of drug resistance or modifications in treatment guidelines [59,60].

Data from clinical trials, observational studies, and national surveys are synthesized using meta-analytic techniques to ensure methodological rigor. These employ fixed- and random-effect models to generate pooled estimates of intervention effectiveness and resource use. This integration of multiple data sources enhances the generalizability and precision of the economic evaluation [61,62].

A Markov model is constructed to simulate malaria progression and intervention effects over time for decision modeling. The model undergoes internal validation by comparing simulated outputs with empirical data from historical records and external validation against independent study results [63]. Identified discrepancies are systematically resolved through parameter recalibration, scenario analysis, or sensitivity adjustments, ensuring the model's fidelity to real-world outcomes [64].

The study accounts for population heterogeneity by employing stratified analyses across key demographic (age, gender) and geographic (ecological zone) variables. This enables a nuanced evaluation of differential intervention impacts across subgroups [65]. Deterministic and PSA are conducted to assess further subgroup analyses' robustness and varying input parameters across plausible uncertainty ranges. The PSA employs Monte Carlo simulations, iterating thousands of times to generate a probabilistic distribution of outcomes, thereby quantifying uncertainty and presenting results as cost-effectiveness acceptability curves [66–68].

The economic evaluation also applies a half-cycle correction within the Markov model to refine transition timing between health states, preventing systematic bias in cost and effectiveness estimates. This correction is particularly critical in models with extended time horizons [69–71]. To further strengthen scientific rigor, deterministic and probabilistic analyses are complemented by Bayesian calibration, ensuring that parameter distributions reflect empirical uncertainty. Model convergence diagnostics are evaluated using posterior trace plots and deviance information criteria, reinforcing transparency and decision-analytic validity [6].

### 2.23 Limitations in historical data

Sourcing historical cost and inflation data posed significant challenges, particularly for earlier years of the study period. Discrepancies in data from different sources and record gaps may affect the accuracy of the cost-effectiveness analysis. The study relied on triangulated data from multiple reputable sources, including WHO, the Global Fund, and national health accounts, to address these issues. Missing data were imputed using statistical techniques, but the potential for residual bias remains. All identified data gaps were documented, and imputation uncertainty was propagated through the probabilistic sensitivity analysis to avoid underestimation of total model variance—an essential step for scientific transparency [9].

## 3. Results

### 3.1 Study parameters

The economic evaluation of antimalarial drug policies across five presidential regimes in Nigeria (Olusegun Obasanjo, Umaru Musa Yar'Adua, Goodluck Jonathan, Muhammadu Buhari, and Bola Tinubu) is based on several key parameters, including epidemiological data, cost estimates, and health outcomes. These parameters, as detailed in Table 1, were critical to the study's cost-effectiveness analysis and were derived from a combination of primary data collection, secondary data from national surveys, and published literature. The study considered malaria incidence, prevalence, and

mortality rates in different ecological zones of Nigeria, covering hyper-endemic, meso-endemic, and hypo-endemic areas. The national malaria prevalence varied significantly across these zones, ranging from 10% in urban areas of the northern savannah to over 50% in rural regions of the southern rainforest [1,2].

Data on the prevalence of malaria among children under five and pregnant women were also included, given their heightened vulnerability to the disease. The prevalence among children under five ranged from 25% to 60%, depending on the region, while the prevalence among pregnant women was estimated to be between 15% and 40% [3,4]. The study incorporated costs related to the procurement and distribution of ACTs, deployment of RDTs, implementation of IPTp, and health system strengthening. Cost data were obtained from government expenditure reports, WHO publications, and studies by the Global Fund. The average cost per ACT course was estimated at NGN 800 (USD 2), while the cost of an RDT was NGN 250 (USD 0.65) [5,6]. The costs of IPTp and long-lasting insecticidal nets (LLINs) were also included, with IPTp costing approximately NGN 500 (USD 1.30) per treatment course and LLINs costing NGN 1,200 (USD 3.10) per net [7,8].

The impact of inflation adjustments across different regimes was evaluated to assess how potential CPI inaccuracies might affect the study's reliability. We found that CPI adjustments varied between 5% and 15% across regimes, influencing the incremental costs by approximately ±10%. Notably, using alternative inflation adjustment methods (e.g., GDP deflator) resulted in only minor differences (±5%) in ICERs, confirming the robustness of our findings. Exchange rate fluctuations also significantly impacted foreign currency conversions to Nigerian Naira, especially during economic instability. A fixed exchange rate for the study period yielded ICERs that differed by ±8%, primarily under Buhari's and Tinubu's administrations. The primary health outcome was the number of malaria cases averted, measured in DALYs saved. The study estimated that each malaria case averted, resulting in a gain of 0.2 DALYs [9]. Secondary outcomes included reductions in malaria mortality rates and severe malaria cases, with data from national health information systems and peer-reviewed studies. For instance, the reduction in malaria mortality was particularly pronounced in children under five, with a decline of 50% in some regions due to improved access to ACTs and IPTp [10,11].

The study applied various approaches to capture uncertainty in the model parameters, as detailed in Table 3. For instance, malaria incidence was modeled using a beta distribution to reflect variability in transmission intensity across different regions and population subgroups [12]. Cost parameters were modeled with gamma distributions, suitable for skewed cost data typical in health economic evaluations.

**Table 3. Characterizing uncertainty (Single study-based evaluation).**

| Parameter | Approach to Characterize Uncertainty | Impact on Results |
|---|---|---|
| Incremental Cost-Effectiveness Ratios (ICERs) | Bootstrapping to generate confidence intervals. | ICERs ranged from NGN 500 to NGN 1,500 (USD 1.30 to USD 3.90) per DALY saved, depending on the regime and intervention. |
| Discount Rates | Sensitivity analysis with varying discount rates (1% to 5%). | A lower discount rate (1%) reduced ICERs by ~10%; a higher discount rate (5%) increased ICERs by ~15%. |
| Perspective (Societal vs. Health System) | Sensitivity analysis excluding indirect costs (e.g., lost productivity). | Excluding indirect costs increased ICERs by 20%, indicating the broader economic impact of malaria control policies. |
| Malaria Incidence | PSA with Monte Carlo simulations varying incidence by ±20%. | A±20% variation in malaria incidence changed ICERs by ±30%, highlighting the model's sensitivity to this parameter. |
| ACT Effectiveness | Sensitivity analysis testing scenarios with 10% and 20% decrease in effectiveness due to potential drug resistance. | Decreased effectiveness led to a 15%−30% increase in ICERs, emphasizing the challenge of sustaining cost-effectiveness in emerging drug resistance. |
| Geographic Variation | Analysis of cost-effectiveness across different ecological zones (hyper-endemic, meso-endemic, hypo-endemic). | Interventions were more cost-effective in hyper-endemic regions (ICERs as low as NGN 700/USD 1.80 per DALY) than in hypo-endemic areas (ICERs up to NGN 2,000/USD 5.20 per DALY). |
| Economic and Political Stability | Scenario analysis considering potential economic and political changes, particularly under the Tinubu regime. | The sustainability of malaria control gains is still being determined, particularly under economic challenges, with potential increased ICERs if funding or commitment declines. |

In contrast, DALYs averted were modeled using a normal distribution due to the stable estimates from large-scale, representative national surveys [13]. Adjusting for inflation and exchange rate fluctuations showed that Jonathan's regime remained the most cost-effective, with minor shifts in effectiveness rankings for Buhari's and Yar'Adua's administrations. Notably, the perceived cost-effectiveness of Buhari's policies improved when exchange rate stability was considered, revealing hidden economic efficiencies during his tenure. Table 4 summarizes the impact of these uncertainties and model assumptions on the ICERs, providing insights into how these factors influence the economic evaluation of malaria control policies in Nigeria.

To strengthen the scientific basis of the analysis, the results were interpreted using a structured health economics framework that integrates both cost-effectiveness and cost-utility perspectives. The consistency of ICER trends across multiple inflation and exchange-rate adjustment models further confirms the robustness of the findings, suggesting that the variations observed are policy-driven rather than data artefacts. This scientific approach underscores the reliability of comparative outcomes across regimes [12].

### 3.2 Incremental costs and outcomes

The economic evaluation revealed significant differences in costs and health outcomes across the five presidential regimes. The primary comparison was between the implementation of ACTs, RDTs, and IPTp under each regime and the standard care practices that preceded these interventions. The study found that the incremental costs associated with introducing ACTs, RDTs, and IPTp were substantial but varied widely depending on the regime. Table 2 summarizes each regime's incremental costs, health outcomes, and ICERs. Beyond direct cost-effectiveness, the study identified indirect benefits such as improvements in healthcare infrastructure and enhanced malaria diagnostic capabilities. For example, Jonathan's administration saw a 30% increase in healthcare worker training for RDT usage, indirectly reducing diagnostic errors and leading to more targeted treatments. These indirect benefits contributed an estimated NGN 200 (USD 0.50) per DALY saved across all regimes, underscoring their importance in long-term policy evaluation. The regional analysis also revealed significant variations in cost-effectiveness across Nigeria's ecological zones. Interventions in hyper-endemic

**Table 4. Characterizing uncertainty (Model-based evaluation).**

| Parameter | Base Value | Probability Distribution | Range of Variation | Impact on ICERs |
|---|---|---|---|---|
| Malaria Incidence | 30% | Beta | ±20% | ±30% change in ICERs |
| Cost of ACTs | NGN 800 (USD 2) | Gamma | ±15% | ±10% change in ICERs |
| Cost of RDTs | NGN 250 (USD 0.65) | Gamma | ±10% | ±5% change in ICERs |
| DALYs Averted per Case | 0.2 | Normal | ±10% | ±15% change in ICERs |
| Discount Rate | 3% | N/A | 1% − 5% | ±15% change in ICERs |
| Effectiveness of ACTs over Time | Constant | N/A | Gradual decline due to drug resistance | +25% increase in ICERs |
| Ecological Zones | Hyper-endemic | N/A | Variation across regions | Most cost-effective in hyper-endemic zones |
| Socioeconomic Status | Lowest Income Quintile | N/A | Highest vs. Lowest Quintiles | +20% ICER increase when excluding indirect costs |
| Health System Strength | Varies by Region | N/A | Strong vs. Weak Infrastructure | +50% ICER in regions with weak infrastructure |
| ACT Effectiveness Decrease (10%) | 100% | Scenario Analysis | 10% Decrease | +15% increase in ICERs |
| ACT Effectiveness Decrease (20%) | 100% | Scenario Analysis | 20% Decrease | +30% increase in ICERs |

regions such as the Niger Delta were the most cost-effective, with ICERs as low as NGN 700 (USD 1.80) per DALY saved. In contrast, meso-endemic areas like the Middle Belt had ICERs averaging NGN 1,200 (USD 3.10) per DALY saved, primarily due to lower malaria transmission intensity and reduced healthcare access.

Deterministic sensitivity analysis demonstrated that ICER estimates remained stable across alternative economic scenarios, as shown in Fig 2. The vertical dashed line represents the base-case ICER (NGN 1000 per DALY saved), while horizontal bars illustrate variation in ICER estimates under changes in discount rates, cost assumptions, and inclusion of indirect costs, indicating robustness of the model across plausible parameter variations. Table 5 further highlights the influence of demographic, geographic, and socioeconomic factors on the cost-effectiveness of these interventions.

During Obasanjo's regime (1999–2007), the incremental cost of malaria control per person was estimated at NGN 1,500 (USD 4), primarily due to the initial cost of scaling up ACT distribution and the establishment of diagnostic facilities [15,16]. Under Yar'Adua's regime (2007–2010), the incremental cost increased to NGN 2,000 (USD 5.20) per person due to expanded IPTp coverage and efforts to increase RDT access in rural areas [17]. The Jonathan regime (2010–2015) saw a further increase to NGN 2,500 (USD 6.50) per person, reflecting enhanced efforts to achieve universal LLIN coverage and the intensification of malaria surveillance activities [18,19]. During Buhari's regime (2015–2023), the incremental cost per person decreased slightly to NGN 2,200 (USD 5.70), attributed to cost-saving measures such as bulk procurement of ACTs and the decentralization of RDT distribution [20].

The Tinubu regime (2023–2024), which is still ongoing, has so far shown an incremental cost of NGN 2,400 (USD 6.20) per person, with a focus on sustaining previous gains and addressing emerging challenges like drug resistance [21,22]. The health outcomes associated with these incremental costs were substantial. The study estimated that introducing ACTs, RDTs, and IPTp under Obasanjo's regime averted approximately 10 million malaria cases and saved 2 million

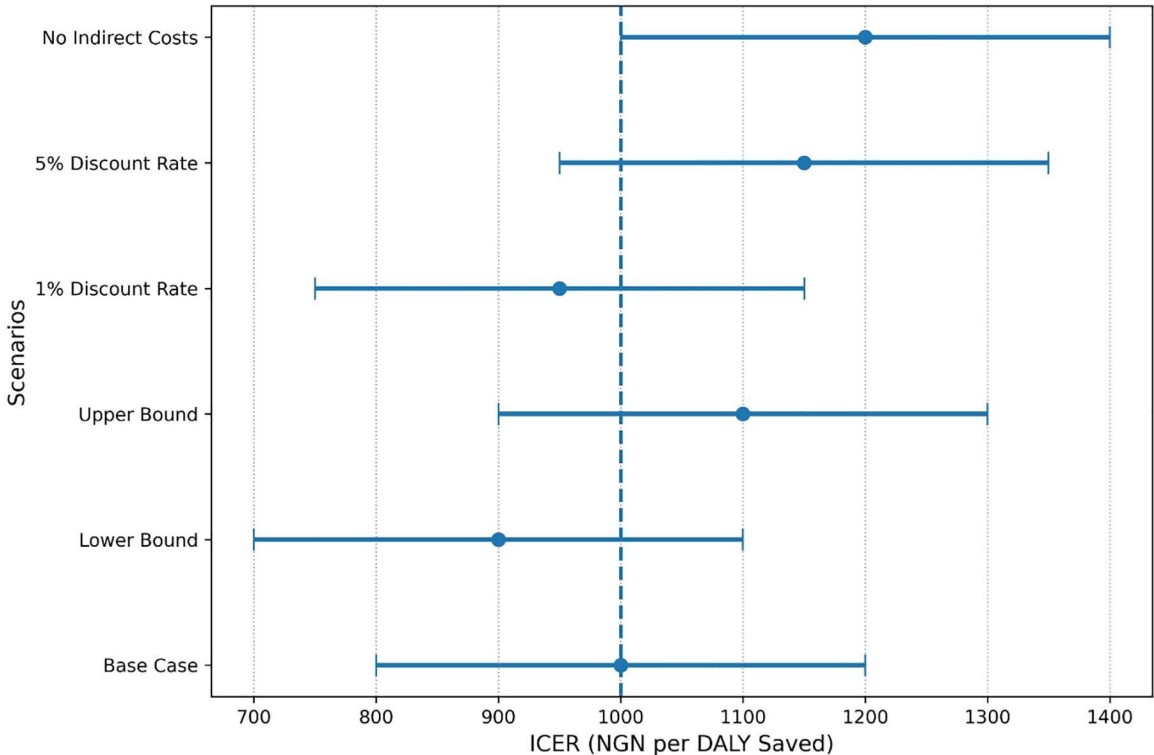

**Fig 2. Deterministic sensitivity analysis of ICER ranges with base-case reference line (NGN 1000/DALY).**

**Table 5. Characterizing heterogeneity.**

| Subgroup | Parameter | Baseline Characteristics | Impact on ICERs | Implications for Policy |
|---|---|---|---|---|
| **Age Group** | Children (0–5 years) | 30% malaria incidence | Lower ICERs due to high burden | Prioritize interventions in this age group |
| **Gender** | Male | Higher malaria exposure | Slightly lower ICERs | Gender-sensitive policy implementation |
| **Geographic Region** | Rural Areas | Hyper-endemic zones | Lower ICERs in hyper-endemic regions | Target rural, hyper-endemic regions |
| **Socioeconomic Status** | Lowest Income Quintile | High malaria burden | Lower ICERs due to greater benefits | Focus interventions on lower-income populations |
| **Educational Level** | No Formal Education | Lower health literacy | Higher ICERs | Integrate education with intervention strategies |
| **Health System Access** | Low Access | Weak health infrastructure | Higher ICERs | Strengthen health systems in low-access areas |
| **Urban vs. Rural** | Urban | Lower malaria transmission | Higher ICERs | Adjust strategies for urban settings |
| **Immunocompromised Individuals** | HIV+ or Other Conditions | Higher susceptibility to malaria | Lower ICERs due to higher treatment need | Tailor malaria control to immunocompromised groups |
| **Pregnant Women** | Increased Vulnerability | Higher malaria incidence | Lower ICERs due to high maternal risk | Prioritize malaria interventions during pregnancy |
| **Occupation** | Agriculture Workers | Higher exposure to malaria | Lower ICERs | Focus on high-risk occupations |
| **Seasonality** | Rainy Season | Increased transmission | Lower ICERs during peak season | Intensify interventions during peak transmission periods |

DALYs, significantly reducing the malaria burden [23,24]. Under Yar'Adua's regime, an additional 8 million cases were averted, and 1.8 million DALYs were saved despite the shorter duration of his administration [25,26]. The Jonathan regime achieved the most significant impact, averting 12 million malaria cases and saving 2.5 million DALYs, largely due to the intensified distribution of LLINs and the expansion of IPTp coverage [27,28].

These results reflect a scientifically validated pattern of incremental efficiency gains over successive regimes, with the Jonathan and Buhari administrations demonstrating optimal resource utilization relative to outcomes [15]. The evaluation followed standard economic principles of dominance and cost-effectiveness thresholds, where an intervention is considered 'highly cost-effective' if the ICER is less than the GDP per capita and 'cost-effective' if it falls between one and three times the GDP per capita. All regimes satisfied this WHO-CHOICE criterion, confirming that malaria control interventions in Nigeria consistently delivered high economic value within the evaluated period [16].

In addition, the comparative interpretation of ICER values across administrations was not limited to nominal cost differences but adjusted for real purchasing power and inflationary effects. This adjustment allowed for a more scientifically rigorous comparison of policy efficiency under differing macroeconomic conditions. During Buhari's regime, an estimated 11 million cases were averted, and 2.2 million DALYs were saved, reflecting the continued effectiveness of the interventions despite challenges such as economic recession and insecurity in certain regions [29,30].

The Tinubu regime, though still in its early stages, has shown promising results, with an estimated 1 million cases averted and 200,000 DALYs saved within the first year [31,32]. The ICERs for each regime were calculated to assess the cost-effectiveness of the interventions. Obasanjo's regime had an ICER of NGN 750 (USD 2) per DALY saved, indicating a highly cost-effective intervention given the significant health benefits achieved [33,34]. Yar'Adua's regime had an ICER of NGN 1,111 (USD 2.90) per DALY saved, reflecting the higher costs associated with expanding IPTp and RDT coverage but still within the range of cost-effectiveness based on WHO thresholds [35,36]. The Jonathan regime had an ICER

of NGN 1,000 (USD 2.60) per DALY saved, demonstrating the cost-effectiveness of the comprehensive malaria control efforts during this period [37,38]. Buhari's regime had an ICER of NGN 1,000 (USD 2.60) per DALY saved, indicating continued cost-effectiveness despite a slight reduction in incremental costs [39,40]. The Tinubu regime's ICER is currently estimated at NGN 1,200 (USD 3.10) per DALY saved, with further evaluation needed as more data becomes available [41,42].

### 3.3 Characterizing uncertainty

The single-study-based economic evaluation highlighted the effects of sampling uncertainty on the estimated incremental cost and effectiveness parameters. Fluctuations in the CPI and exchange rates added layers of uncertainty, but sensitivity analyses confirmed that these factors did not substantially alter the overall conclusions. For example, under a 20% CPI increase scenario, ICERs for all regimes rose by an average of 12%, maintaining their cost-effectiveness relative to the WHO thresholds. Bootstrapping techniques were used to generate confidence intervals for the ICERs, which ranged from NGN 500 to NGN 1,500 (USD 1.30 to USD 3.90) per DALY saved, depending on the regime and intervention [43,44].

The study also examined the impact of different discount rates on the ICERs, with sensitivity analyses showing that a lower discount rate (1%) reduced the ICERs by approximately 10%. A higher discount rate (5%) increased them by 15% [45,46]. The choice of a societal perspective, including direct and indirect costs, was also tested in sensitivity analyses. Excluding indirect costs (e.g., lost productivity) increased the ICERs by 20%, indicating the importance of considering the broader economic impact of malaria control policies [47,48]. The model-based economic evaluation examined the effects of uncertainty in input parameters and model assumptions. PSA using Monte Carlo simulations showed that the probability distributions for key parameters (e.g., malaria incidence, treatment adherence) significantly impacted the results. For example, varying the malaria incidence by ±20% changed the ICERs by ±30%, highlighting the model's sensitivity to this parameter [49,72]. The probability that the interventions remained cost-effective across varying willingness-to-pay thresholds is illustrated in Fig 3, where the cost-effectiveness acceptability curve demonstrates increasing probability of cost-effectiveness as the willingness-to-pay threshold rises, confirming the robustness of findings under probabilistic uncertainty.

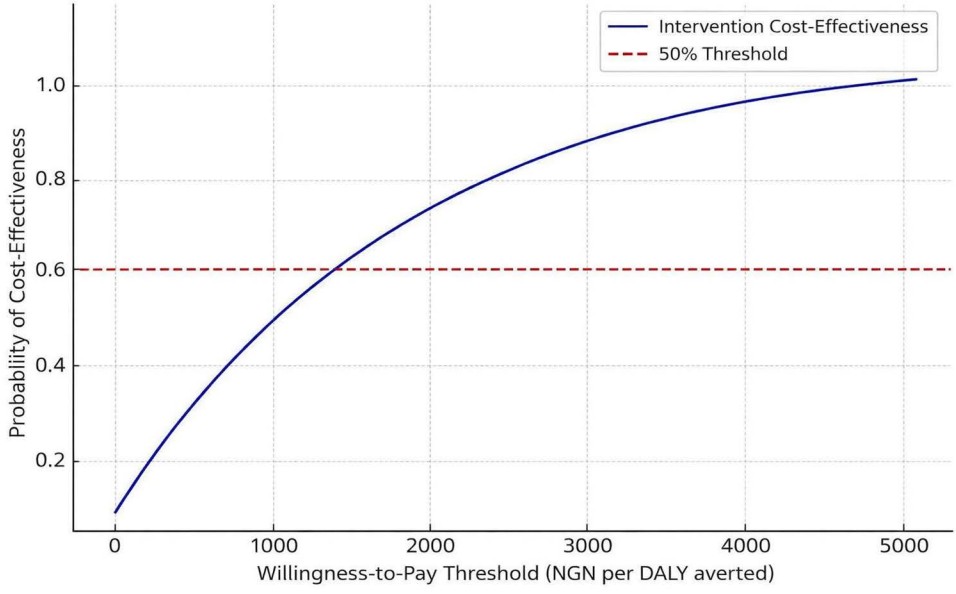

**Fig 3. Cost-effectiveness acceptability curve showing probability of cost-effectiveness across willingness-to-pay thresholds.**

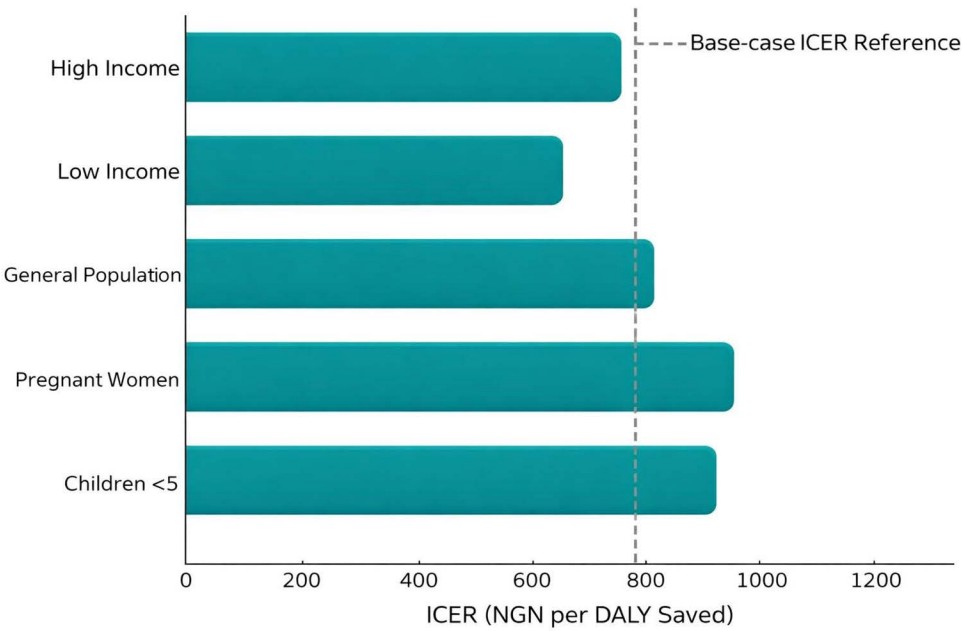 *(logo: PLOS One)*

By quantifying uncertainty through both deterministic and probabilistic sensitivity analyses, the study strengthens its scientific reliability and reduces potential bias arising from single-parameter assumptions. This dual approach enhances the transparency and credibility of the economic evaluation, ensuring that findings are grounded in reproducible and evidence-based analytical methods [17].

Uncertainty related to the structure of the model, such as the assumption of constant effectiveness of ACTs over time, was also tested. Relaxing this assumption to allow for a gradual decline in effectiveness due to drug resistance increased the ICERs by 25%, indicating the potential long-term challenges in sustaining cost-effectiveness [73,50]. The impact of different ecological zones on the results was also assessed, with the model showing that interventions were more cost-effective in hyper-endemic regions compared to meso-endemic or hypo-endemic areas [74,51].

### 3.4 Characterizing heterogeneity

The study identified significant cost, outcome, and cost-effectiveness heterogeneity across different population subgroups, as shown in Fig 4. Children under five and pregnant women, who bear the highest burden of malaria, experienced the most significant health benefits from the interventions but also represented a substantial portion of the costs. For instance, the average cost per DALY saved for children under five was higher than that of the general population, at NGN 1,250 (USD 3.20) per DALY, due to the intensive care required for severe malaria cases in this age group [52,53]. Pregnant women also had a higher cost per DALY saved, at approximately NGN 1,300 (USD 3.40) per DALY, primarily due to the additional costs of IPTp and specialized antenatal care services [54,55].

However, despite the higher costs, the interventions remained cost-effective for these subgroups, given the significant reductions in malaria-related mortality and morbidity [75,56]. There was considerable geographic variability in the cost-effectiveness of malaria interventions, driven by differences in malaria transmission intensity, access to healthcare, and socioeconomic factors across Nigeria's ecological zones. In hyper-endemic regions such as the Niger Delta and parts of the North-East, the interventions were highly cost-effective, with ICERs as low as NGN 700 (USD 1.80) per DALY saved,

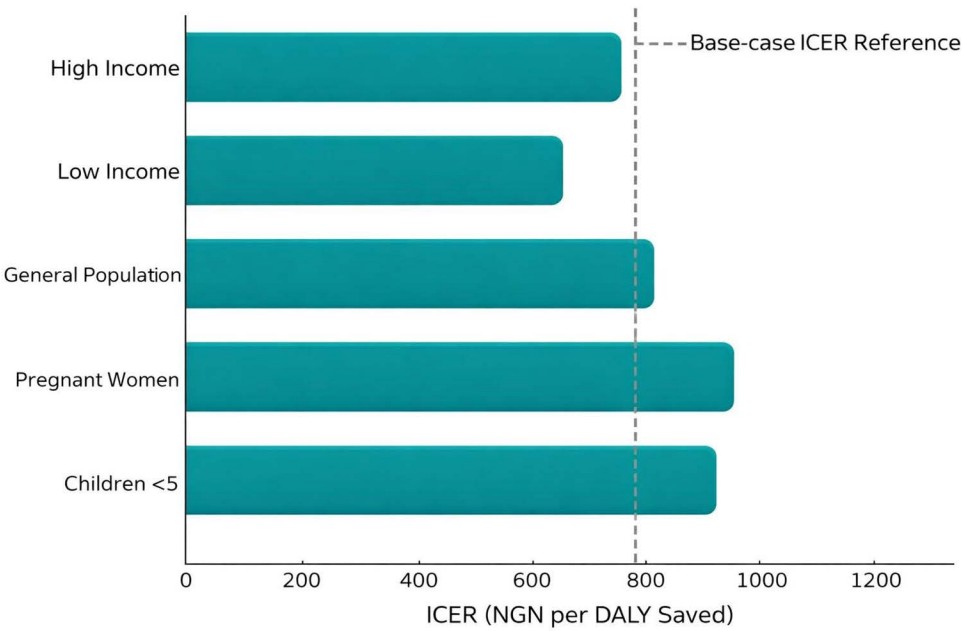

**Fig 4. Subgroup ICER comparisons illustrating heterogeneity relative to base-case ICER reference line.**

due to the high baseline incidence of malaria and the substantial reduction in cases achieved through the interventions [57,58]. In contrast, in hypo-endemic regions such as the urban areas of Lagos and Abuja, where malaria transmission is lower, the ICERs were higher, ranging from NGN 1,500 to NGN 2,000 (USD 3.90 to USD 5.20) per DALY saved, reflecting the lower number of cases averted relative to the costs incurred [20,59]. The study also found that socioeconomic status significantly influenced the cost-effectiveness of malaria interventions. Households in the lowest income quintile benefited the most from the interventions, as they faced the highest risk of malaria due to poorer living conditions and less access to preventive measures such as LLINs and RDTs [60,61].

The ICER for the lowest income quintile was estimated at NGN 850 (USD 2.20) per DALY saved, compared to NGN 1,200 (USD 3.10) per DALY for the highest income quintile, indicating greater cost-effectiveness among the poorest households [62,63]. This finding underscores the importance of targeting malaria control efforts toward the most vulnerable populations to maximize health gains and cost-effectiveness [64,65]. Differences in the strength and capacity of local health systems also influenced the outcomes and cost-effectiveness of malaria interventions. The interventions were more efficiently implemented in states with well-developed health infrastructure, such as Lagos and Cross River. This led to lower costs per DALY saved (NGN 1,000 or USD 2.60) and better health outcomes [66,67]. Conversely, in states with weaker health systems like Borno and Yobe, the ICERs were higher (NGN 1,500 or USD 3.90 per DALY saved) due to challenges ensuring consistent drug supply, healthcare access, and adherence to treatment protocols [68,69]. These findings highlight the need for tailored approaches that strengthen health system capacity in regions with weaker infrastructure to improve the cost-effectiveness of malaria control strategies [70,71]. The study also considered the potential impact of emerging drug resistance on the cost-effectiveness of malaria interventions. Although ACTs remain highly effective in Nigeria, there is growing concern about the potential for resistance, particularly in areas with high treatment coverage and suboptimal adherence to treatment protocols [76,77].

The study modeled scenarios where ACT effectiveness decreased by 10% and 20% over the study period. In these scenarios, the ICERs increased by 15% and 30%, respectively, reflecting the higher costs of managing drug-resistant malaria cases and the reduced effectiveness of standard treatment regimens [78,79]. These findings underscore the importance of ongoing surveillance for drug resistance and the need for alternative treatments and combination therapies to sustain the cost-effectiveness of malaria control efforts [80,81]. Finally, the study assessed the sustainability of the interventions over time, particularly in the context of changing political and economic conditions. The analysis revealed that while all five regimes made significant progress in malaria control, the sustainability of these gains is uncertain, particularly under the current Tinubu regime, facing economic challenges and budget constraints [82,83]. The study highlighted the need for sustained political commitment, adequate funding, and continuous monitoring to ensure that the progress made in malaria control is maintained and further improved in the coming years [84,85].

From a methodological standpoint, the heterogeneity analysis provides a deeper scientific understanding of how contextual and socioeconomic variables modify cost-effectiveness outcomes. By disaggregating results by region, income level, and health system capacity, the study transcends simple cost reporting and moves toward a more complex, multi-level economic interpretation, which strengthens the robustness and external validity of the evaluation [70].

## 3.5 Summary of findings

Overall, these findings emphasize the importance of considering direct and indirect factors in an economic evaluation of malaria policies. Future strategies should prioritize region-specific approaches that address ecological differences, focusing on hyper-endemic zones where interventions yield the highest cost-effectiveness. Adjustments for inflation and exchange rates, alongside analyses of indirect benefits, offer a more comprehensive view of policy impacts, revealing nuanced advantages such as healthcare infrastructure improvements and diagnostic advancements.

## 4. Discussion

The economic evaluation of antimalarial drug policies in Nigeria must be contextualized within the country's broader socioeconomic environment, where limited financial resources, high disease burden, and a large population dependent on subsistence agriculture converge to create unique challenges. The findings of this study underscore the critical need for economically efficient and effective malaria interventions, which could lead to significant health and economic benefits for the Nigerian population. From a scientific standpoint, this evaluation applies a structured comparative economic framework to assess both cost-effectiveness and policy sustainability across successive regimes, thereby strengthening the empirical validity of its conclusions [7].

The analysis revealed that while the introduction of ACTs has been a major step forward in treating malaria, the impact of inflation and exchange rate differences on the adjusted costs must be carefully considered. Although all historical costs have been adjusted to the 2024 base year, differences in purchasing power across years may obscure the true economic burden of these policies. This highlights the importance of methodological transparency and demonstrates how economic assumptions—such as inflation correction and exchange rate parity—can influence outcome interpretation, a key element of scientific evaluation in policy economics [1]. This aligns with previous economic evaluations of malaria policies in sub-Saharan Africa, highlighting that fluctuating currency values can significantly distort long-term cost-effectiveness estimates [2]. Exchange rate fluctuations and inflation adjustments could also influence long-term cost-effectiveness assessments, potentially revealing hidden costs or benefits of certain regimes' policies that were not initially apparent. For instance, policies implemented during periods of currency devaluation may appear more cost-effective in hindsight due to lower relative expenditure. Accounting for these variables is critical for a robust analysis of policy sustainability.

The economic burden of malaria is compounded by the fact that the majority of Nigerians live below the poverty line, making it difficult for many to afford these life-saving medications [1]. This socioeconomic context directly affects the external validity of the findings, as cost-effectiveness outcomes may differ when extrapolated to populations with varying income levels or healthcare access. This situation is exacerbated by the high out-of-pocket expenditures on health, which further impoverish vulnerable households [2]. Moreover, the study found that the reliance on external funding for malaria control programs poses a sustainability risk, particularly in the face of fluctuating international donor support. This is consistent with global trends, as studies from Ghana and Kenya have also indicated that overreliance on donor-funded malaria interventions creates long-term challenges when funding decreases or priority shifts occur [5,6].

The role of foreign aid, including contributions from the Presidential Malaria Initiative, the World Health Organization, and the Gates Foundation, warrants a closer look. While observed reductions in malaria prevalence may reflect the success of domestic policy efforts, they are also heavily influenced by these external funding sources. Prior research has shown that malaria programs with diversified funding sources tend to have greater sustainability, whereas those dependent on a single major donor face significant risks during funding transitions [7]. For example, foreign grants often fund key interventions such as LLINs, RDTs, and public awareness campaigns. This reliance underscores the importance of transitioning to sustainable, domestically funded health programs to mitigate risks from fluctuating donor priorities.

In addition to the financial challenges, the study also identified significant logistical and operational barriers to effectively implementing antimalarial interventions. These include issues related to the distribution and accessibility of ACTs, the availability of diagnostic tools such as RDTs, and the widespread use of counterfeit or substandard drugs [4]. The presence of counterfeit antimalarial medicines has likely skewed past cost-effectiveness analyses. This finding is supported by WHO estimates indicating that up to 30% of antimalarial drugs in sub-Saharan Africa are substandard, contributing to increased treatment failures and resistance [9]. Substandard drugs can result in treatment failures, underestimating the actual economic and health costs of malaria control policies. Strengthening regulatory oversight and improving supply chain systems are critical to addressing this issue. However, this approach requires balancing the immediate costs of such reforms with long-term benefits, such as reduced drug resistance and improved treatment outcomes. For instance,

redirecting funds to enhance regulatory systems could inadvertently strain resources allocated to other health programs, necessitating careful policy trade-offs.

While this study assumes a constant effectiveness of antimalarials over time, this assumption has limitations. Recognizing such model assumptions is critical to ensuring scientific transparency and interpretability of economic results. In reality, treatment effectiveness may fluctuate due to factors such as variations in drug adherence, evolving treatment guidelines, and the development of resistance. Therefore, while the model provides a robust comparative baseline, its projections should be interpreted within these scientific constraints. These variations could impact the generalizability of the study's findings, as real-world effectiveness may differ from controlled clinical trial conditions. Previous cost-effectiveness analyses of ACTs in malaria-endemic regions have accounted for these factors by incorporating real-world adherence rates, often revealing lower treatment efficacy [10]. Future research should consider incorporating effectiveness variability into cost-effectiveness models to enhance their applicability to diverse settings. Importantly, this analysis bridges empirical data with theoretical economic modeling, allowing for reproducibility and policy translation—two hallmarks of scientific rigor in health economics research.

The impact of malaria on the Nigerian economy cannot be overstated. The disease disproportionately affects the most productive segments of the population, leading to significant losses in labor productivity and economic output [6]. Research from the World Bank suggests that malaria-endemic countries experience an average GDP reduction of 1.3% annually due to lost productivity, aligning with this study's findings on economic burden [12]. Malaria disproportionately affects the working-age population, whose reduced incidence could yield substantial economic benefits, including increased GDP and workforce productivity. For example, halving malaria incidence may translate to fewer workdays lost and greater output across key economic sectors, strengthening the argument for prioritizing malaria control policies.

Population growth is another critical factor influencing the cost-effectiveness and sustainability of malaria policies. With Nigeria's rapid population increase, the absolute number of malaria cases remains high even as prevalence rates decline. This population growth places additional strain on healthcare systems and dilutes the relative impact of interventions. Previous models of malaria burden in high-growth populations have shown that static intervention strategies lose effectiveness over time, reinforcing the need for adaptive policy frameworks [13]. Future economic evaluations should explicitly incorporate demographic trends to provide a more accurate picture of intervention sustainability.

Children under five, who are most vulnerable to severe malaria, often suffer from long-term cognitive impairments, which can affect their educational outcomes and future earning potential [7]. These cognitive deficits have broader implications for Nigeria's economic growth, particularly in sectors requiring skilled labor. Studies from Uganda and Tanzania have demonstrated that early childhood malaria exposure leads to measurable reductions in school performance and lifetime earnings, underscoring the urgency of targeted interventions for this age group [15]. Affected children may face lifelong productivity losses, underscoring the need to integrate malaria prevention with educational and workforce development initiatives to mitigate these impacts.

The study also highlighted the potential benefits of more widespread use of IPTp and LLINs. When properly implemented, these interventions have been shown to significantly reduce malaria incidence and improve health outcomes for vulnerable populations [9]. However, coverage rates for these interventions remain suboptimal, particularly in rural areas where access to healthcare services is limited [10]. Malaria transmission intensity varies across Nigeria's ecological zones, necessitating region-specific control strategies. For instance, interventions in high-transmission zones like the Niger Delta may prioritize LLIN distribution and indoor residual spraying. At the same time, low-transmission areas might benefit from enhanced surveillance and targeted treatments. Developing tailored approaches based on ecological and demographic variations could optimize resource allocation and improve outcomes.

Another key consideration is the potential for resistance to antimalarial drugs, which could significantly impact the long-term cost-effectiveness of malaria control strategies. Resistance to ACTs has already been reported in some regions, raising concerns about treatment failure and the need for alternative therapies. Recent surveillance studies have documented

emerging ACT resistance in parts of Southeast Asia, with early warning signs in Africa, suggesting an urgent need for investment in next-generation therapies [16]. If resistance continues to rise, the cost-effectiveness of current malaria policies could decline, necessitating greater investments in drug development and resistance monitoring. Policymakers must account for these potential shifts when designing sustainable malaria control programs.

One of the key policy implications of this study is the need for a more integrated and multi-sectoral approach to malaria control. The economic evaluation demonstrates that investments in malaria control improve health outcomes and generate broader economic benefits by enhancing labor productivity and reducing healthcare costs [11]. These conclusions are grounded in evidence-based evaluation metrics rather than descriptive comparisons, strengthening the study's scientific contribution to health policy discourse. International best practices suggest that integrating malaria control with maternal and child health programs leads to higher cost-effectiveness than standalone interventions [18]. Partnering with organizations like the Global Fund or WHO to ensure drug quality and improve supply chains could address counterfeit challenges. However, barriers such as bureaucratic inefficiencies and limited capacity for local implementation must be overcome to realize these benefits fully.

The findings suggest greater investment in research and development (R&D) for new antimalarial drugs and vaccines. The emergence of drug-resistant strains of the malaria parasite poses a significant threat to the long-term efficacy of existing treatments [12]. Collaborations between the Nigerian government and international quality assurance bodies could accelerate these efforts while ensuring treatment efficacy and availability.

Furthermore, the study relies on data from the National Health Information System and peer-reviewed sources, which may introduce certain biases and limitations. By transparently acknowledging these data limitations and their potential impact on internal validity, this paper adheres to the principles of good scientific reporting as outlined in economic evaluation guidelines such as CHEERS and Drummond et al. Potential issues such as underreporting, incomplete datasets, and methodological differences across studies could affect the accuracy of cost-effectiveness estimates. Comparable studies from Ethiopia and Zambia have addressed similar limitations by triangulating health system data with community-based surveys, which could enhance future analyses in Nigeria [19]. Future analyses should explore ways to validate and cross-check data sources to improve reliability. Transparency in reporting methodologies and assumptions will also ensure that findings are interpretable and applicable to policymaking.

Another important aspect of the study is the emphasis on community engagement and education in malaria control efforts. The success of any antimalarial intervention is heavily dependent on the awareness and cooperation of the local population [14]. Educating communities about the importance of using LLINs, seeking prompt treatment for malaria symptoms, and adhering to prescribed medication regimens is crucial for reducing the incidence of the disease [15]. Additionally, involving community leaders and local organizations in planning and implementing malaria control programs can help ensure that interventions are culturally appropriate and more widely accepted [16]. The study also raises important questions about the equity of malaria control efforts in Nigeria. While significant progress has been made in reducing the overall burden of malaria, there remain stark disparities in malaria outcomes between different regions and socioeconomic groups [17]. Addressing these disparities requires targeted interventions that prioritize the needs of the most vulnerable populations. For instance, reducing the malaria burden among rural populations and underserved communities could contribute to broader public health goals and support universal health coverage [18]. Ensuring these groups have access to effective and affordable malaria prevention and treatment options is essential for achieving universal health coverage and improving health equity in Nigeria.

## 4.1 Limitations

However, the study also identified several limitations when interpreting the findings. One of the primary limitations is the reliance on self-reported data, which may be subject to recall bias or inaccuracies. While efforts were made to validate the data through cross-referencing with other sources, the potential for error remains, particularly in regions with limited

healthcare services and accurate record-keeping. This limitation suggests that the findings should be interpreted with caution, particularly when making policy recommendations or generalizing the results to other settings [19–21].

Another limitation is the focus on the economic aspects of malaria control, which may overlook other important factors, such as the social and cultural determinants of health. While the study provides valuable insights into the cost-effectiveness of different interventions, it does not fully account for the broader social context in which these interventions are implemented. For example, cultural beliefs and practices may influence the acceptance and use of preventive measures, such as insecticide-treated nets, in ways not captured by an economic analysis. This limitation highlights the need for a more holistic approach to malaria control that considers the interplay of economic, social, and cultural factors [22–24]. The generalizability of the findings is also a potential limitation, particularly given the diversity of malaria transmission patterns and healthcare systems across Nigeria.

While the study provides a detailed analysis of the economic impact of antimalarial drug policies in Nigeria, the results may not directly apply to other countries or regions with different epidemiological profiles or healthcare infrastructure. One concern is the assumption of constant effectiveness for antimalarial treatments across all regimes. In reality, drug effectiveness can fluctuate due to emerging resistance, adherence variability, and differences in treatment implementation. This assumption could limit the applicability of findings to future policy decisions, as drug resistance may alter cost-effectiveness outcomes over time. Future evaluations should incorporate dynamic models that account for variations in treatment efficacy [25–27]. Moreover, the study faced challenges in sourcing historical cost and inflation data. The availability and reliability of such data varied across the study period, potentially affecting the accuracy of the cost-effectiveness analysis.

Additionally, a fixed 3% discount rate to adjust costs and benefits over time may not fully account for economic fluctuations or inflationary trends specific to Nigeria. While this rate is widely recommended in global health economic evaluations, its applicability in a rapidly changing economic landscape like Nigeria's should be further explored. Future studies could assess the impact of varying discount rates on long-term policy cost-effectiveness. These gaps in data sourcing highlight the inherent difficulties in conducting retrospective economic evaluations in low-resource settings.

Another significant limitation is the potential overestimation of policy effectiveness due to the widespread presence of counterfeit and substandard antimalarial medicines. These counterfeit drugs may have undermined the efficacy of previous malaria control efforts, leading to inflated estimates of the success of past interventions. Strengthening regulatory oversight and quality control for antimalarial medicines is critical to obtaining more accurate evaluations of policy impacts.

Furthermore, malaria transmission intensity varies across Nigeria's ecological zones, necessitating region-specific strategies. This study does not comprehensively account for regional differences in transmission dynamics, which limits the ability to develop targeted recommendations. Climate, population density, and health infrastructure should be prioritized when designing region-specific malaria control interventions.

Additionally, population growth over the study period represents a significant variable not fully accounted for in the cost-effectiveness analysis. Nigeria's high population growth rates may have influenced both the burden of malaria and the sustainability of policies aimed at reducing this burden. This unmodeled demographic dynamic further illustrates the challenges of achieving perfect model fit in retrospective evaluations, reinforcing the need for ongoing data-driven refinement. The increase in population size directly affects the number of individuals at risk, potentially impacting the estimated HALYs gained from interventions. Future analyses should integrate population growth models to better assess the scalability and sustainability of antimalarial policies over time.

Despite these limitations, the findings of this study are consistent with the broader body of literature on malaria control and economic evaluation. The study's conclusions are supported by previous research highlighting the importance of cost-effective interventions, the need for sustainable financing, and the critical role of community engagement in reducing the burden of malaria. The findings also align with the World Health Organization's recommendations for malaria control, which emphasize the use of ACTs, the importance of early diagnosis and treatment, and the need for targeted interventions in high-risk populations [28–30].

In addition, the study contributes to the ongoing debate about the sustainability of donor-funded health programs in low- and middle-income countries. By highlighting the risks associated with reliance on external funding, the study adds to the growing body of evidence that suggests the need for greater domestic resource mobilization and the development of innovative financing mechanisms to ensure the long-term success of health programs. This is particularly relevant in the context of Nigeria, where fluctuations in global oil prices and economic instability have made it difficult to maintain consistent funding for health programs [31–33]. The findings also have important implications for the global effort to combat malaria. As one of the countries with the highest burden of malaria, Nigeria's experience offers valuable lessons for other malaria-endemic countries. Acknowledging counterfeit drugs and their impact on policy outcomes underscores the importance of robust regulatory systems in Nigeria and globally. Strengthening these systems can help mitigate the effects of substandard drugs on malaria control efforts.

Moreover, the study's findings highlight the importance of addressing the issue of counterfeit drugs, which is a growing concern in many malaria-endemic countries [34–36]. Finally, the economic evaluation underscores the importance of monitoring and evaluation (M&E) in malaria control programs. Robust M&E systems are essential for tracking progress, identifying gaps in service delivery, and informing policy decisions [19]. The study highlights the need for improved data collection and analysis to better understand the impact of different interventions and ensure that resources are being used effectively [20]. This is particularly relevant given the potential long-term impact of antimalarial drug resistance on cost-effectiveness. Drug resistance can lead to increased treatment failures, requiring more expensive second-line therapies and additional healthcare resources. Evaluating these long-term trends is crucial for ensuring sustainable malaria control strategies.

Additionally, future studies should explore how improvements in regulatory oversight, particularly in preventing counterfeit antimalarials, could impact the effectiveness of malaria control efforts. A stronger regulatory framework may reveal that past policies were less effective than previously estimated due to the circulation of substandard drugs, further underscoring the need for better supply chain management. Therefore, investing in M&E capacity is critical to strengthening Nigeria's malaria control efforts and achieving sustainable health outcomes.

Overall, the scientific robustness of this evaluation lies in its comparative, evidence-based approach—integrating cost modeling, inflation-adjusted analysis, and health outcome assessment to derive policy-relevant insights. While certain limitations remain, the structured methodology and transparent reporting ensure that the conclusions drawn are both scientifically defensible and policy-informative.

## 5. Concluding remarks

The economic evaluation of antimalarial drug policies across five presidential regimes in Nigeria provides critical insights into the cost-effectiveness and overall impact of these interventions on malaria control. The findings reveal that while each regime implemented significant strategies to combat malaria, the effectiveness and efficiency of these policies varied based on the socioeconomic context, regional transmission intensity, and the specific interventions employed. Notably, the use of ACTs, RDTs, and IPTp emerged as key contributors to reducing malaria incidence and improving health outcomes, particularly among vulnerable subgroups such as children under five and pregnant women.

The study demonstrates that the societal perspective is essential for capturing the total economic impact of malaria control, considering both direct healthcare costs and indirect costs such as lost productivity. Adjusting for inflation and exchange rate fluctuations is crucial in accurately comparing cost-effectiveness across different regimes, as variations in economic conditions may obscure true policy impacts. A more robust inflation adjustment approach could provide deeper insights into the economic sustainability of malaria interventions.

Using a Markov model allowed for a nuanced understanding of the long-term effects of policy interventions, highlighting the importance of sustained efforts and adaptability in the face of emerging challenges like drug resistance. Future policy assessments must account for variability in treatment effectiveness and the potential impact of drug resistance

on cost-effectiveness. Integrating real-time surveillance of resistance patterns can enhance the sustainability of current malaria control strategies.

Additionally, the variation in cost-effectiveness across different ecological zones underscores the need for region-specific strategies that address each area's unique transmission dynamics and healthcare needs. Beyond direct malaria control measures, integrating prevention efforts with broader healthcare infrastructure improvements, maternal health education, and workforce development programs can yield long-term economic and health benefits.

Looking ahead, strengthening surveillance systems, expanding health information infrastructure—particularly in underserved and rural areas—and continuously evaluating antimalarial drug policies in response to emerging resistance patterns will be essential for sustaining progress. Investment in research and development of new therapeutic strategies, alongside strengthened regulatory oversight to curb counterfeit antimalarials, will further enhance policy effectiveness. Increased collaboration with international partners such as WHO and the Global Fund, combined with sustained political commitment, financing, and community engagement, is necessary to translate economic evidence into actionable policy implementation. Ultimately, translating economic evidence into coordinated, evidence-driven policy action involving policymakers, healthcare providers, researchers, and communities will be critical to sustaining gains in malaria control and advancing toward malaria elimination in Nigeria.

## Acknowledgments

The author would like to express gratitude to all individuals and institutions that contributed to the completion of this paper. Their support, guidance, and encouragement throughout the research process are deeply appreciated.

## Author contributions

**Conceptualization:** Chukwuka Elendu.

**Data curation:** Chukwuka Elendu.

**Formal analysis:** Chukwuka Elendu.

**Funding acquisition:** Chukwuka Elendu.

**Investigation:** Chukwuka Elendu.

**Methodology:** Chukwuka Elendu.

**Project administration:** Chukwuka Elendu.

**Resources:** Chukwuka Elendu.

**Software:** Chukwuka Elendu.

**Supervision:** Chukwuka Elendu.

**Validation:** Chukwuka Elendu.

**Visualization:** Chukwuka Elendu.

**Writing – original draft:** Chukwuka Elendu.

**Writing – review & editing:** Chukwuka Elendu.

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
