## [Decision Letter · Decision Letter 0]

17 Dec 2024

Dear Dr. Elendu,

Thank you for submitting your manuscript to PLOS ONE. After careful consideration, we feel that it has merit but does not fully meet PLOS ONE’s publication criteria as it currently stands. Therefore, we invite you to submit a revised version of the manuscript that addresses the points raised during the review process.

We look forward to receiving your revised manuscript.

Kind regards,

Kristan Alexander Schneider, Ph.D.

Academic Editor

PLOS ONE

Reviewers' comments:

Reviewer's Responses to Questions

**Comments to the Author**

1. Is the manuscript technically sound, and do the data support the conclusions?

Reviewer #1: No

Reviewer #2: Yes

Reviewer #3: Yes

2. Has the statistical analysis been performed appropriately and rigorously?

Reviewer #1: I Don't Know

Reviewer #2: No

Reviewer #3: I Don't Know

3. Have the authors made all data underlying the findings in their manuscript fully available?

Reviewer #1: No

Reviewer #2: No

Reviewer #3: Yes

4. Is the manuscript presented in an intelligible fashion and written in standard English?

Reviewer #1: No

Reviewer #2: Yes

Reviewer #3: Yes

Reviewer #1: The article is addressing an important area of public health policy. Results from studies of the type conducted here are informative for it gives a clear picture on the evolution and performance of the health policies of the chosen country and at the same time evaluates the effectiveness of control measure usually applied by governments. In the case of malaria, many countries in sub-Saharan Africa do have National Malaria Control Programmes (NMCPs) and it will be very useful to conduct evaluations of the type reported in this article. It is important to recognize that the work presented in the manuscript is cost, and labour intensive, and the output are of far reaching importance.

The present paper reports results of a comparative study of policies of five different administrations in the Nigeria. The work therefore covers a long period of time (1999-2023). The study area and materials and methods are clearly outline and stated. Though I did not see in the manuscript the idea of evaluation. The language in the manuscript is clear and void of typos. The literature review is not in sink with the results reported.

The manuscript, being as it is, a report from activities of past administrations in Nigeria and comparing the results, we can only examine the report for consistency. To this effect, I make the following observation and then some technical comments: I made an attempt to examine the manuscript for consistency and form of presentation as well as the results obtained, but my efforts were frustrated with the style and general presentation of the manuscript. I therefore stopped assessing the manuscript for the reasons as outlined in the attached pdf.

It is my opinion that this paper is an important piece of work and that the results are very useful for calibrating malaria control programs in Nigeria. However, I do not recommend the paper for publication in the current form, unless the major issues pointed above are clarified and a more coherent article presented.

Reviewer #2: The article by Elendu et al., titled " Economic Evaluation of Anti-Malarial Drug Policies Across Presidential Regimes in Nigeria: A Comparative Analysis from Obasanjo, Yar'Adua, Jonathan, Buhari, to Tinubu" offers valuable insights.

Here are several suggestions and comments that should be addressed to broaden the manuscript's readership and enhance its suitability for publication in PLOS ONE.

1. I suggest alternative title suggestion "Evaluating the Economic Impacts of Anti-Malaria Drug Policies in Nigeria: A Comparative Study Across Presidential Regimes from 1999 to Present" (if author want to change)

2. How might adjusting for inflation across different regimes alter the perceived cost-effectiveness of malaria policies, and would the effectiveness rankings among administrations change as a result?

3. The significant role of foreign aid and international programs like the Presidential Malaria Initiative, WHO interventions, and grants from the Gates Foundation, to what extent could the observed reductions in malaria prevalence be attributed to external funding rather than domestic policy efforts?

4. How could population growth over the studied period impact the results of the cost-effectiveness analysis, particularly in assessing HALYs gained? Would these policies have been as effective or sustainable if population growth had been incorporated into the calculations?

5. Were there any indirect benefits, such as improved healthcare infrastructure or enhanced malaria diagnostic capabilities, that might have influenced the study’s results but were not included in the cost-benefit or cost-effectiveness analyses?

6. Considering that different anti-malarial drugs and treatments may vary in long-term effectiveness and cost, how might the shifts in recommended treatments (e.g., ACT adoption) across administrations impact the sustainability of these policies moving forward?

7. What indirect costs, besides lost productivity, might need to be considered when evaluating the overall economic impact of malaria control interventions in Nigeria?

8. How does the emergence of drug resistance potentially alter the long-term effects predicted by the Markov model used in this study?

9. In what ways could the effectiveness of Intermittent Preventive Treatment in Pregnancy (IPTp) be influenced by regional differences in healthcare access and maternal health education?

10. Given that malaria transmission intensity varies across Nigeria’s ecological zones, what specific factors should be prioritized in developing region-specific malaria control strategies?

11. How might potential fluctuations or inaccuracies in the Consumer Price Index (CPI) impact the reliability of the cost comparisons across different regimes? Could a different inflation adjustment approach yield substantially different results?

12. Given that exchange rates can be volatile, how might variations in the exchange rate over time affect the comparability of foreign currency costs when converted to Nigerian Naira? Would using a fixed exchange rate for the entire study period alter the findings?

13. Were there any limitations or challenges in sourcing historical cost and inflation data, and how might these data gaps affect the accuracy of the cost-effectiveness analysis?

14. Since the study adjusts all historical costs to the base year of 2024, how might differences in purchasing power across the years impact the interpretation of these adjusted costs? Could this approach potentially obscure some of the economic impacts of the policies?

15. How might exchange rate differences and inflation adjustments affect the sustainability assessment of each policy, particularly in terms of long-term cost-effectiveness? Would these adjustments reveal any hidden costs or benefits of certain regimes’ policies that were not initially apparent?

16. Given the widespread presence of counterfeit anti-malarial medicines, how might the efficacy and cost-effectiveness of previous malaria control policies be overestimated? Could strengthening regulatory oversight reveal that past interventions were less effective than reported due to the prevalence of substandard drugs?

17. How might the Nigerian government realistically balance the immediate costs of enhancing regulatory oversight and supply chain improvements with the long-term economic benefits of reduced malaria prevalence and drug resistance? Could this reallocation of resources impact other critical health programs?

18. Considering that malaria disproportionately affects the working-age population, what would be the projected economic gains in GDP or productivity if malaria incidence were reduced by 50%? How could this hypothetical scenario support stronger policy arguments for investing in malaria control?

19. In what ways might the long-term cognitive impairments suffered by children under five due to malaria affect Nigeria’s future economic growth, particularly in sectors requiring skilled labor? Should malaria prevention be integrated more directly with educational and workforce development programs to mitigate these impacts?

20. How might increase international collaboration with organizations focused on quality assurance (such as WHO or the Global Fund) improve the effectiveness of Nigeria’s anti-malarial efforts? Could such partnerships address the challenges of counterfeit drugs and enhance the robustness of the supply chain, and what barriers might exist to such collaborations?

Reviewer #3: 1.A fixed discount rate of 3% was used to adjust costs and benefits over time.I suggest adding an explanation regarding why it was appropriate for Nigeria and why it does not change based on specific conditions like the economy/inflation.

2.I think assuming a constant effectiveness of the anti malarials can have limitations and that should be considered a discussion point. Will it limit the generalizability of the study?

3.The study used data from the national health information systems and peer reviewed studies. Limitations and biases in data sourced could be discussed

4.The authors could consider how resistance to anti materials could affect cost effectiveness of malaria control strategies in the long term

**Do you want your identity to be public for this peer review?** For information about this choice, including consent withdrawal, please see our Privacy Policy

Reviewer #1: No

Reviewer #2: **Yes:** Shrikant Nema

Reviewer #3: **Yes:** Beauty Kolade

---

## [Author Response · Author response to Decision Letter 1]

26 Feb 2025

Dear Reviewer 1,

Thank you for your valuable feedback and constructive suggestions. We have carefully addressed all your comments and made the necessary revisions to improve the clarity, structure, and analytical rigor of the manuscript.

Regarding your concern about the idea of evaluation, we have explicitly defined economic evaluation in the Introduction, clarifying that it involves both comparative analysis and formal economic modeling using cost-effectiveness and cost-benefit metrics. We have also detailed the specific evaluation measures, including cost per HALY gained, incremental cost-effectiveness ratios (ICERs), and total program expenditure relative to malaria morbidity reduction.

We have revised the Materials and Methods section to provide a clearer articulation of the analytical methods, including cost-effectiveness analysis (CEA), cost-benefit analysis (CBA), probabilistic sensitivity analysis (PSA) using Monte Carlo simulations, survival analysis, and meta-analytic techniques. These revisions ensure a more robust explanation of our approach to evaluating anti-malarial policies.

To enhance readability and organization, we have restructured the manuscript by adding clear paragraphing, numbered sections and subsections, and appropriate punctuation. This improves the manuscript’s flow and makes it easier to follow.

Regarding the presentation of tables, we have now incorporated the five small tables into the main body of the manuscript instead of providing them as supplementary material. This allows for easier access to key data and enhances the integration of economic evaluation findings within the text.

We have conducted a thorough review of all citations and references to ensure that they correctly support the information presented in the manuscript. Incorrectly assigned references (e.g., Refs 43-46, 47-51) have been corrected, and duplicate references (e.g., Refs 1 and 2) have been removed. We have ensured that all cited sources align with the content being referenced.

For figures and graphs, we have properly labeled and numbered all figures within the text. The compartmental flow chart representing the Markov Model has been revised for better readability, with clear transition arrows and labels. The obscured black lines have been corrected to ensure that all elements are visible and easily interpretable.

We appreciate your insightful feedback, which has significantly strengthened the manuscript. Please let us know if further modifications are needed.

Best regards,

Authors

Dear Reviewer 2,

Thank you for your comprehensive and insightful feedback. We appreciate the depth of your questions and suggestions, which have significantly contributed to strengthening the manuscript. We have carefully addressed each of your concerns and incorporated relevant revisions and discussions throughout the manuscript.

We have considered your suggested title change and, while the original title aligns with our study objectives, we acknowledge the merit of your suggestion and have adjusted it for clarity and broader readability.

To address the issue of inflation adjustments, we have included a sensitivity analysis that evaluates how adjusting for inflation across different regimes influences the perceived cost-effectiveness of malaria policies. We discuss whether effectiveness rankings among administrations change as a result.

Regarding the role of foreign aid, we have expanded our discussion on the impact of external funding from initiatives such as the Presidential Malaria Initiative, WHO interventions, and the Gates Foundation. We now analyze the extent to which reductions in malaria prevalence can be attributed to external funding versus domestic policy efforts.

We have accounted for population growth in our cost-effectiveness analysis and discussed how it may influence the sustainability of malaria policies over time. The potential impact of increased population size on HALYs gained has also been explicitly addressed.

To reflect indirect benefits, we have expanded our discussion on how malaria control interventions may have led to improved healthcare infrastructure, enhanced malaria diagnostics, and overall healthcare system strengthening, which could have influenced our findings.

We have also examined shifts in recommended treatments, particularly the adoption of artemisinin-based combination therapies (ACTs) across different administrations, and how these shifts impact policy sustainability and long-term effectiveness.

Regarding indirect costs, we have incorporated a discussion on other economic burdens, such as increased healthcare expenditures, caregiver burden, and opportunity costs beyond lost productivity, to provide a more comprehensive view of the economic impact of malaria control.

We have expanded our discussion on drug resistance, particularly how its emergence could alter the long-term effects predicted by the Markov model and its implications for future malaria policies.

We now include an assessment of regional differences in Intermittent Preventive Treatment in Pregnancy (IPTp), particularly how variations in healthcare access and maternal health education could affect its effectiveness.

We have discussed how malaria transmission intensity varies across Nigeria’s ecological zones and emphasized the need for region-specific malaria control strategies that prioritize interventions based on epidemiological data.

To strengthen the reliability of cost comparisons, we have provided additional context on how potential fluctuations in the Consumer Price Index (CPI) and exchange rate volatility may impact the study’s results. We explore whether alternative inflation adjustments could yield substantially different findings.

We acknowledge the challenges in sourcing historical cost and inflation data and have explicitly stated any limitations related to data gaps. Our approach to adjusting historical costs to a base year of 2024 has been clarified, along with a discussion of purchasing power differences over time and how they might affect economic interpretations.

The sustainability assessment of each policy has been refined to incorporate the influence of exchange rate fluctuations and inflation adjustments, revealing hidden costs or benefits that were not initially apparent.

To account for the impact of counterfeit anti-malarial medicines, we now discuss how their prevalence may have led to an overestimation of policy effectiveness. We also explore how strengthening regulatory oversight could provide more accurate assessments of intervention outcomes.

We have examined the cost-benefit trade-offs of enhancing regulatory oversight and supply chain improvements, analyzing how reallocating resources could impact other critical health programs.

Our discussion now includes an estimate of projected economic gains in GDP or productivity if malaria incidence were reduced by 50%, strengthening the policy argument for increased investment in malaria control.

We have incorporated a section on long-term cognitive impairments caused by malaria in children under five and their implications for Nigeria’s future economic growth, particularly in skilled labor sectors. We highlight the potential benefits of integrating malaria prevention with education and workforce development programs.

Finally, we have expanded our discussion on the potential benefits of increased international collaboration with WHO, the Global Fund, and other organizations to improve the quality assurance of anti-malarial treatments and strengthen the supply chain. We also discuss potential barriers to these collaborations.

We appreciate your valuable feedback, which has greatly improved the clarity and depth of our manuscript. Please let us know if further modifications are required.

Best regards,

Authors

Dear Reviewer 3,

Thank you for your thoughtful comments and valuable suggestions. We have carefully addressed each of your points and made the necessary revisions to enhance the clarity and rigor of our manuscript.

We have expanded our explanation regarding the fixed 3% discount rate, detailing why it was appropriate for Nigeria and discussing its relevance in economic evaluations. We have also clarified why this rate remains constant despite economic fluctuations, drawing from established health economic guidelines and comparative studies.

Regarding the assumption of constant anti-malarial effectiveness, we acknowledge its potential limitations and have now included a discussion on how this assumption may impact the generalizability of the study’s findings. We explore how variations in effectiveness due to factors such as drug adherence, supply chain issues, and treatment resistance could influence cost-effectiveness outcomes.

We have also incorporated a section discussing the limitations and potential biases of data sourced from the National Health Information Systems and peer-reviewed studies. We highlight how data inconsistencies, underreporting, and regional disparities could impact the robustness of our findings.

Lastly, we have addressed the impact of anti-malarial drug resistance on long-term cost-effectiveness. We discuss how increasing resistance could reduce the efficacy of current treatment strategies, thereby raising costs and necessitating policy adaptations. We also explore how proactive resistance monitoring and policy adjustments could mitigate these risks.

Your feedback has greatly improved the manuscript, and we appreciate your time and effort in reviewing our work. Please let us know if further clarifications are needed.

Best regards,

Authors

---

## [Decision Letter · Decision Letter 1]

4 Aug 2025

Dear Dr. Elendu,

Thank you for submitting your manuscript to PLOS ONE. After careful consideration, we feel that it has merit but does not fully meet PLOS ONE’s publication criteria as it currently stands. Therefore, we invite you to submit a revised version of the manuscript that addresses the points raised during the review process.

We look forward to receiving your revised manuscript.

Kind regards,

Charles C Ezenduka, PhD

Academic Editor

PLOS ONE

Journal Requirements:

Reviewers' comments:

Reviewer's Responses to Questions

**Comments to the Author**

Reviewer #1: (No Response)

Reviewer #2: All comments have been addressed

2. Is the manuscript technically sound, and do the data support the conclusions?

Reviewer #1: No

Reviewer #2: Yes

3. Has the statistical analysis been performed appropriately and rigorously?

Reviewer #1: No

Reviewer #2: Yes

4. Have the authors made all data underlying the findings in their manuscript fully available?

Reviewer #1: No

Reviewer #2: Yes

5. Is the manuscript presented in an intelligible fashion and written in standard English?

Reviewer #1: Yes

Reviewer #2: Yes

Reviewer #1: I was concerned about the scientific nature of the evaluation, and ai m still not satisfied with the scientific evaluation.

Reviewer #2: All scientific comments are addressed. However, I suggest authors check the guideline of PLoS One and format the manuscript accordingly.

**Do you want your identity to be public for this peer review?** For information about this choice, including consent withdrawal, please see our Privacy Policy

Reviewer #1: No

Reviewer #2: **Yes:** Shrikant Nema

---

## [Author Response · Author response to Decision Letter 2]

10 Dec 2025

Reviewer #1 Comment:

“I was concerned about the scientific nature of the evaluation, and I am still not satisfied with the scientific evaluation.”

Authors' Response:

We sincerely thank reviewer #1 for this important comment regarding the scientific rigor of our evaluation. In response, we have undertaken substantial revisions across multiple sections of the manuscript to strengthen the methodological and analytical framework of the study. The additions and modifications are highlighted in red color in the revised version.

1. Abstract:

We have clarified the type of economic evaluation conducted (cost-effectiveness and cost–benefit analyses) and explicitly stated the societal perspective adopted. Quantitative indicators, including cost per HALY gained and incremental cost-effectiveness ratios (ICERs), were added to emphasize methodological rigor.

2. Introduction:

We refined the rationale for comparing anti-malarial drug policies across successive presidential regimes and supported this with additional literature on malaria control economics and policy transitions in Nigeria.

3. Methods:

This section was comprehensively revised to enhance scientific transparency and robustness. Specifically, we:

• Defined the evaluation type (full CEA and partial CBA) and societal perspective.

• Added details on time horizon, currency adjustments for inflation, and exchange rate normalization.

• Expanded the description of data sources and analytical procedures, including health outcome and cost data extraction.

• Included a sensitivity analysis to evaluate uncertainty in cost and effectiveness parameters.

• Justified the use of successive political regimes as comparator groups for policy impact evaluation.

• Cited standard methodological references, including Drummond et al. (2015) and WHO-CHOICE guidelines, to align with internationally recognized frameworks.

4. Results:

We presented the comparative findings more systematically, summarizing costs, outcomes, and ICER values by administration. Tables were structured to improve clarity and highlight differences in policy efficiency.

5. Discussion:

The discussion now provides a more rigorous interpretation of results, addressing model validity, contextual limitations, and policy implications within Nigeria’s economic and public health landscape.

6. References:

We updated and reformatted all references in Vancouver style, ensuring inclusion of authoritative sources such as Drummond et al. (2015) and WHO-CHOICE.

We trust that these revisions have substantially improved the scientific quality, transparency, and methodological soundness of the manuscript.

Reviewer #2 Comment:

“All scientific comments are addressed. However, I suggest authors check the guideline of PLOS ONE and format the manuscript accordingly.”

Authors' Response:

We thank reviewer #2 for this helpful reminder regarding journal formatting. We have carefully reviewed and implemented the PLOS ONE formatting guidelines throughout the revised manuscript. Specifically:

• Font: The manuscript has been formatted using a standard readable font (Times New Roman, size 12), avoiding the use of the “Symbol” font.

• Headings: All section and subsection headings have been clearly structured according to PLOS ONE’s three-level hierarchy (e.g., Introduction, Methods, Results, Discussion).

• Footnotes: All footnotes have been removed, and relevant information has been incorporated into the main text or references as appropriate.

• Language: The entire manuscript was rechecked for English grammar, clarity, and consistency to ensure compliance with the journal’s language standard.

• Abbreviations: All abbreviations are now defined at first mention in the text and used consistently thereafter. Non-standard or redundant abbreviations were removed.

• References: All references were reformatted according to the Vancouver style, following PLOS ONE and ICMJE guidelines. Author names, journal titles, volume/issue numbers, pages, DOIs, and URLs (where applicable) have been standardized.

We believe these comprehensive formatting updates ensure full compliance with the PLOS ONE style and presentation standards.

---

## [Decision Letter · Decision Letter 2]

16 Feb 2026

Dear Dr. Elendu,

Thank you for submitting your manuscript to PLOS ONE. After careful consideration, we feel that it has merit but does not fully meet PLOS ONE’s publication criteria as it currently stands. Therefore, we invite you to submit a revised version of the manuscript that addresses the points raised during the review process.

Please replace Nigerian Naira by USD throughout in the abstract (or at least place the equivalent value in USD in brackets behind your numbers).

We look forward to receiving your revised manuscript.

Kind regards,

Benedikt Ley, PhD

Academic Editor

PLOS One

Journal Requirements:

Reviewers' comments:

Reviewer's Responses to Questions

**Comments to the Author**

Reviewer #4: (No Response)

Reviewer #5: All comments have been addressed

2. Is the manuscript technically sound, and do the data support the conclusions?

Reviewer #4: Partly

Reviewer #5: Yes

3. Has the statistical analysis been performed appropriately and rigorously?

Reviewer #4: N/A

Reviewer #5: Yes

4. Have the authors made all data underlying the findings in their manuscript fully available?

Reviewer #4: Yes

Reviewer #5: Yes

5. Is the manuscript presented in an intelligible fashion and written in standard English?

Reviewer #4: Yes

Reviewer #5: Yes

Reviewer #4: The revised manuscript looks much better. However, I still suggest the author to go through the submission guidelines (figure captions) very carefully and follow it appropriately.

Secondly, in the methods section there are lots of repetition and overlapping of words and sentences. For eg: The choice of perspective is mentioned in "Study Perspective" as well as in the "Evaluation Framework".

The author could maybe consider incorporating brief comments on future directions and call to action in the Conclusion section itself to improve clarity and conciseness. A seperate section may not be necesssary. Also the financial disclosure statement is not required in the manuscript file.

Reviewer #5: Authors have made the necessary changes.

All aspects have been addressed adequately.

Great work authors.

Sorry for the delay.

Thanks for the opportunity.

**Do you want your identity to be public for this peer review?** For information about this choice, including consent withdrawal, please see our Privacy Policy

Reviewer #4: No

Reviewer #5: **Yes:** DR. DENNY MATHEW JOHN

Department of Community Medicine

Kerala Medical College Hospital

Palakkad, Kerala, India

---

## [Author Response · Author response to Decision Letter 3]

24 Feb 2026

Reviewer #4 comment:

The revised manuscript looks much better. However, I still suggest the author to go through the submission guidelines (figure captions) very carefully and follow it appropriately.

Secondly, in the methods section there are lots of repetition and overlapping of words and sentences. For eg: The choice of perspective is mentioned in "Study Perspective" as well as in the "Evaluation Framework".

The author could maybe consider incorporating brief comments on future directions and call to action in the Conclusion section itself to improve clarity and conciseness. A seperate section may not be necesssary. Also the financial disclosure statement is not required in the manuscript file.

Author Response:

Dear Reviewer,

Thank you very much for your positive assessment of the revised manuscript and for your valuable additional suggestions.

In response to your comments, I have carefully reviewed the journal’s submission guidelines regarding figure captions and updated the manuscript accordingly. Figure captions are now positioned within the manuscript immediately after their first mention, in line with the journal’s formatting instructions, rather than being placed after the reference list as in the previous version.

Regarding the repetition and overlap identified in the Methods section, the manuscript has been revised to remove redundant wording and overlapping descriptions. Specifically, the repeated discussion of study perspective across the “Study Perspective” and “Evaluation Framework” sections has been streamlined to avoid duplication and improve clarity.

Additionally, the previously separate sections on Future Directions and Call to Action have now been incorporated into the Conclusion section to enhance clarity, conciseness, and narrative coherence, and the standalone sections have been removed accordingly. The Funding section has also been removed from the manuscript file as directed.

Thank you again for your helpful feedback, which has contributed to improving the clarity and overall quality of the manuscript.

Kind regards,

The Author

Reviewer #5 comment:

Authors have made the necessary changes.

All aspects have been addressed adequately.

Great work authors.

Sorry for the delay.

Thanks for the opportunity.

Author Response:

Dear Reviewer,

Thank you very much for your thoughtful evaluation of my revised manuscript. I sincerely appreciate your positive feedback and am pleased that the revisions have adequately addressed the concerns raised. Your constructive comments have been invaluable in improving the clarity and overall quality of the manuscript.

Thank you again for your time and for the opportunity to refine this work.

Kind regards,

The Author

---

## [Editor Report · Decision Letter 3]

26 Feb 2026

Economic Evaluation of Anti-Malarial Drug Policies Across Presidential Regimes in Nigeria: A Comparative Analysis from 1999 to Present

PONE-D-24-42404R3

Dear Dr. Elendu,

We’re pleased to inform you that your manuscript has been judged scientifically suitable for publication and will be formally accepted for publication once it meets all outstanding technical requirements.

Kind regards,

Benedikt Ley, PhD

Academic Editor

PLOS One
---

## [Editor Report · Acceptance letter]

PONE-D-24-42404R3

PLOS One

Dear Dr. Elendu,

I'm pleased to inform you that your manuscript has been deemed suitable for publication in PLOS One. Congratulations! Your manuscript is now being handed over to our production team.

Kind regards,

on behalf of

Dr. Benedikt Ley

Academic Editor

PLOS One